# WebWeaver: Structuring Web-Scale Evidence with Dynamic Outlines for Open-Ended Deep Research

**Zijian Li**[12*]**, Xin Guan**[3]**, Bo Zhang**[2]**, Shen Huang**[2✉]**, Houquan Zhou**[4]**, Shaopeng Lai**[2]**,
**Ming Yan**[2]**, Yong Jiang**[2✉]**, Pengjun Xie**[2]**, Fei Huang**[2]**, Jun Zhang**[1]**, Jingren Zhou**[2]

[1]Hong Kong University of Science and Technology     [2]Tongyi Lab, Alibaba Group
[3]Southeast University     [4]Soochow University
zijian.li@connect.ust.hk, pangda@alibaba-inc.com
eejzhang@ust.hk

## Abstract

This paper tackles **open-ended deep research (OEDR)**, a complex challenge where AI agents must synthesize vast web-scale information into insightful reports. Current approaches are plagued by dual-fold limitations: static research pipelines that decouple planning from evidence acquisition and monolithic generation paradigms that include redundant, irrelevant evidence, suffering from hallucination issues and low citation accuracy. To address these challenges, we introduce **WebWeaver**, a novel dual-agent framework that emulates the human research process. The planner operates in a dynamic cycle, iteratively interleaving evidence acquisition with outline optimization to produce a comprehensive, citation-grounded outline linking to a memory bank of evidence. The writer then executes a hierarchical retrieval and writing process, composing the report section by section. By performing targeted retrieval of only the necessary evidence from the memory bank via citations for each part, it effectively mitigates long-context issues and citation hallucinations. Our framework establishes a new state-of-the-art across major OEDR benchmarks, including DeepResearch Bench, DeepConsult, and DeepResearchGym. These results validate our human-centric, iterative methodology, demonstrating that adaptive planning and focused synthesis are crucial for producing comprehensive, trusted, and well-structured reports. The code has been publicly available at https://github.com/Alibaba-NLP/DeepResearch.

## 1 Introduction

Large Language Models (LLMs) (OpenAI, 2025b; Qwen Team, 2025; Liu et al., 2024; DeepMind, 2025; anthropic, 2025) have demonstrated remarkable capabilities across a wide array of well-defined tasks, from factual question answering (Wei et al., 2025; Mialon et al., 2023) to document summarization (Zhang et al., 2025) and code generation (Jiang et al., 2024). Their success, however, has largely been confined to scenarios with clear instructions and ground-truth answers. The true frontier for autonomous AI lies in transcending these structured problems to tackle the complex, open-ended challenges that define human-level knowledge work—a process driven by curiosity, synthesis, and the discovery of novel insights. We term this challenge open-ended deep research (OEDR). Unlike tasks with ground-truth answers, OEDR requires an agent to independently navigate and digest a vast corpus of information, often exceeding 100 web pages and PDFs, to form a detailed report with accurate citations. This represents a monumental challenge, and as shown in Fig. 1, 2, most proprietary agents fail dramatically on report quality (RACE) and citation accuracy (FACT) of DeepResearch Bench (Du et al., 2025), highlighting a critical gap we aim to address.

Current attempts to tackle OEDR fall into two main categories: proprietary and open-source solutions. While several powerful proprietary agents show strong performance (OpenAI, 2025a; Research,

---

*Work done during an internship at Tongyi Lab, Alibaba Group.
✉Corresponding author.

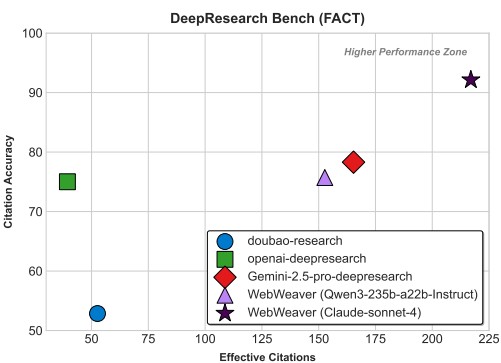

Figure 1: Performance of varying deep research agents on DeepResearch Bench (RACE). The results on DeepResearch Bench are taken from the official leaderboard. Our proposed WebWeaver achieves state-of-the-art performance and even outperforms reference answers.

Figure 2: Performance of varying deep research agents on DeepResearch Bench (FACT). Our proposed WebWeaver achieves highest effective citations and citation accuracy.

2025b;c;a), their prohibitively expensive APIs and restrictive quotas create significant barriers, limiting widespread adoption and hindering academic research. Consequently, the focus has shifted towards open-source alternatives, which predominantly follow two paradigms. As shown in Fig. 3, the first is a straightforward "search-then-generate" approach (Tao et al., 2025; Roucher et al., 2025), where the agent gathers all information before directly generating a report. This method results in low-quality, incoherent outputs because it lacks an outline to guide writing. The second, more sophisticated approach either initializes a static outline to guide searching (Han et al., 2025; Research, 2025d; Researcher, 2025) or searches information for outlining (Shao et al., 2024; Xiong et al., 2025). However, these strategies are critically flawed: the upfront outline often suffers from the LLM's internal outdated knowledge. The search-first method also constrains the searching boundary without outline guidance. Furthermore, feeding all searched materials or redundant evidence into the context for final writing is susceptible to the "loss in the middle" issue (Liu et al., 2023) and increased hallucinations, compromising the report's accuracy and depth (Bai et al., 2024; Wu et al., 2025c).

The key, we believe, lies in abandoning rigid, machine-like pipelines and instead embracing the organic process of human intellect. Our approach is designed to do just that: it teaches the agent to research like a person. A human expert doesn't decouple and fix their drafting and searching phases; they allow these two phases to co-evolve until converging to a comprehensive outline. We implement this principle through an agentic loop where actions of searching and outline optimization interleave. As the agent explores the web-scale information landscape, its discoveries continuously inform and reshape the outline. Critically, this refined outline then acts as a strategic blueprint, actively guiding subsequent searches to fill identified knowledge gaps and explore underdeveloped sections. This creates a true feedback loop where outlining and discovery co-evolve. Then, when it is time to write, our agent should accept only the relevant context. Just as a human writer would refer to specific notes for a specific chapter, our agent composes each section by focusing only on the source-grounded materials. By doing so, it operates with clarity and precision, crafting a final report that is not just a summary of data but well-structured and insightful pieces of analysis with accurate citations.

To this end, we propose **WebWeaver** by following the human-centric philosophy, a dual-agent framework comprising a *planner* and a *writer*. As shown in Fig. 3, the planner embodies the exploratory research phase, operating in a dynamic, agentic cycle that iteratively interleaves evidence acquisition with outline optimization, culminating in a comprehensive, source-grounded research outline, where each section is explicitly linked via citations to a curated memory bank of source evidence. When it turns to the writing phase, to address the critical long context and attentional context management challenge, the writer executes a memory-grounded, citation-driven, hierarchical synthesis process. It constructs the report section by section, performing targeted retrieval of only the relevant evidence from a structured memory bank via citations in the outline for each subtask. This synergistic division of labor enables our agent to navigate complex information landscapes and produce reports that are both comprehensive in scope and trusted in their evidentiary grounding.

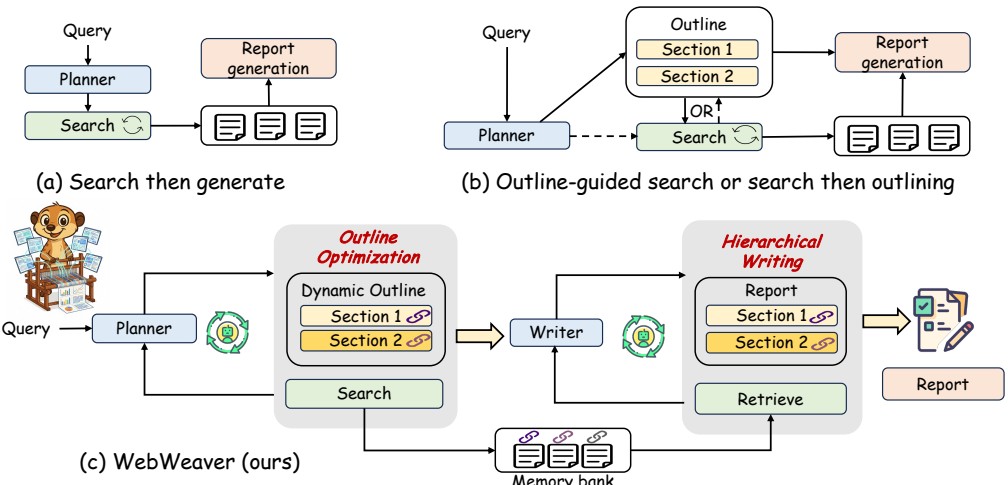

Figure 3: (a) the search-then-generate paradigm first gathers information and then directly generates a report; (b) the paradigms decouple the searching and outline generation; (c) WebWeaver not only enables a dynamic research cycle where the outline and search strategy co-evolve but allows hierarchical and attentional writing by retrieving only relevant evidence with citations in the outline.

Extensive experiments demonstrate that WebWeaver achieves state-of-the-art (SOTA) performance and outperforms both the proprietary and open-source agent systems on three recent and challenging open-ended deep research benchmarks. Detailed discussion is produced to demonstrate the effectiveness of outline optimization and memory-grounded synthesis. Critically, WebWeaver enables agentic finetuning of small models for practical use. We construct a high-quality SFT dataset, WebWeaver-3k, generated by our framework. The experiments with WebWeaver-3k demonstrate that the complex skills of thinking, searching, and writing can be learned, enabling smaller, accessible models to achieve the expert-level performance previously confined to large-scale proprietary systems.

## 2 PRELIMINARIES

**Problem definition.** We consider the open-ended research question without the ground-truth answers. Given an open-ended question, the agents need to search relevant information and finally output a report or article. To achieve this, we implement a planner for collecting information, a memory to store materials, and a writer for report generation. For both the planner and writer, we adopt ReAct (Yao et al., 2023) as the agent's framework. Upon receiving a question, they perform several iterations of *thought-action-observation*. Specifically, in each iteration, based on the existing context, the LLM generates a *thought* and executes a parsable *action*, then awaits the environment to return an *observation*. The planning and writing stages terminate with the output token of "<terminate>". A complete trajectory with $T$ iterations can be defined as

$$\mathcal{H}_T = (\tau_0, a_0, o_0, \ldots, \tau_i, a_i, o_i, \ldots, \tau_T, a_T), \tag{1}$$

where $\tau_i$, $a_i$, $o_i$ represent thought, action, and observation sampled from the planning or writing policy based on all previous context in the $i$-th round, respectively.

**Actions.** For the planner, the action space consists of *search*, *write outline*, and *terminate*. Given the search queries, the search engine returns titles, snippets, and corresponding URLs. To save context space, we further execute the actions of URL selection, parsing pages via URLs, summarizing relevant contents, and extracting evidence with LLMs following the searching queries. The search tool finally returns the selected URLs with their corresponding summaries and evidence. The action of "write outline" is to generate and optimize the outline with citations linking to the evidence in the memory bank, and the "terminate" action is to terminate the planning process.

For the writer, the action space consists of *retrieve*, *write*, and *terminate*. Besides the *terminate* action, the *retrieve* action is to retrieve evidence from the memory bank by providing the grounded citations in the outline. The *write* action is provided to write the section of the report.

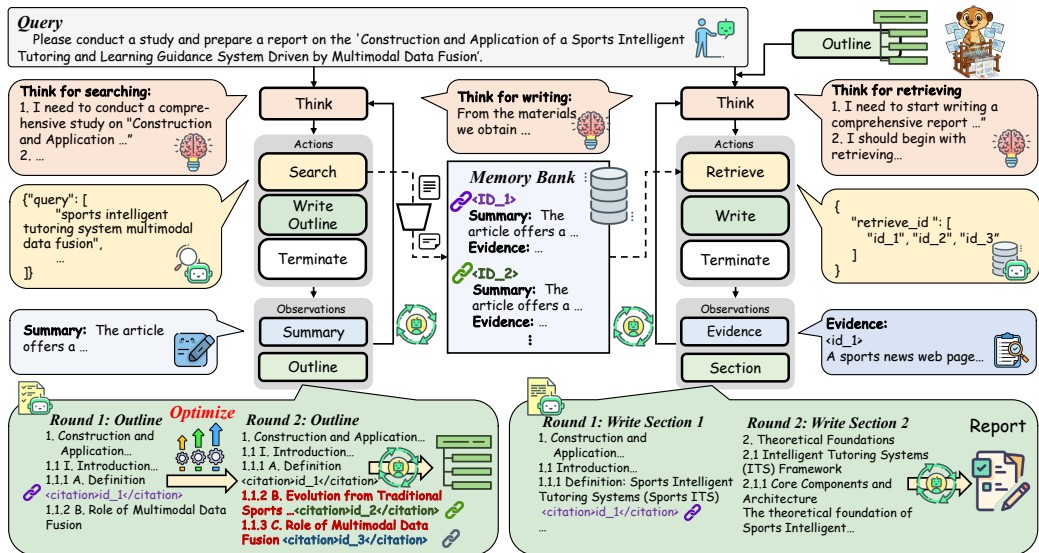

Figure 4: The workflow of WebWeaver. Left: The planner first iteratively collects evidence via the search action and optimizes the outline until outputting a comprehensive and citation-grounded outline. Right: The writer performs hierarchical and attentional writing by retrieving relevant evidence with the grounded citations in the outline.

**Memory bank.** Answering an open-ended question requires long-context input of the collected information and long-context output of the final report. To search sufficient materials, the planner often searches and parses more than 100 web pages, with more than 100k tokens. The writer often outputs more than 20k tokens to produce a comprehensive report. Prior open-sourced deep research agents (Roucher et al., 2025; Research, 2025d; Researcher, 2025) include all the raw materials (e.g., web pages and PDF files) in the LLM context, leading to quality degradation due to attentional failures like the "lost in the middle" problem, poor coherence, and increased hallucinations (Liu et al., 2023; Li et al., 2024a; Bai et al., 2024; Wu et al., 2025c). To this end, we introduce a memory to achieve context management for both planner and writer. Only a short summary of the web page or PDF file is included in the search context, and only necessary raw pages will be retrieved from the memory to write the corresponding sections via the citations in the outline.

## 3 METHOD

Our methodology is embodied in a dual-agent framework, comprising a planner and a writer. The planner is responsible for the dynamic cycle of evidence acquisition and outline optimization, while the writer performs evidence-grounded, section-by-section synthesis to construct the final report. This division of labor directly mirrors the cognitive workflow of a human researcher. The entire workflow is visualized in Fig. 4.

The planner operates in a dynamic, agentic research cycle. It iteratively interleaves evidence acquisition from web searches with the continuous refinement and optimization of a report outline. The output of this exploratory phase is not just a collection of sources but a comprehensive, well-structured outline where each section is explicitly linked via citations to a curated memory bank of source evidence. Subsequently, the writer takes over for the synthesis phase. To circumvent the pitfalls of brute-force generation and noisy-context issues, the writer adopts a section-wise and memory-grounded synthesis approach. For each section of the outline, it performs targeted retrieval of only the pertinent evidence from the memory bank using cations and composes the content. This division of labor ensures that the final report is not only coherent and well-organized but also deeply source-grounded, faithfully mirroring the rigor of human-led deep research.

## 3.1 RESEARCH CYCLE: ITERATIVE EVIDENCE ACQUISITION AND OUTLINE OPTIMIZATION

Recent deep research agents primarily follow two distinct, yet equally flawed, paradigms that decouple planning from discovery. The first, an "outline-guided search" approach (Han et al., 2025; Research, 2025d; Researcher, 2025), generates a static outline before any evidence is gathered. This rigidly confines the research to the LLM's internal, often outdated knowledge, making it blind to emergent insights. The second, a "search-then-outlining" approach (Shao et al., 2024; Xiong et al., 2025), performs an initial, broad search and then generates a fixed outline based on these preliminary findings. This approach permanently constrains the research scope to what was discovered in the initial, undirected search. Despite their different ordering, both paradigms share the same critical limitation: their one-way workflow prevents adaptation. In contrast, we propose a dynamic research cycle where the outline and search strategy co-evolve, creating a feedback loop that allows the planner to continuously adapt and explore in response to new findings.

The core of our planner's operation is a dynamic research cycle that iteratively interleaves evidence acquisition with outline optimization. Unlike static approaches, our planner continuously adapts its strategy based on emergent findings. For each step, the planner selects one of the three actions: search, write outline, and terminate.

**Evidence acquisition.** When the planner thinks that there is still insufficient evidence or knowledge to make a comprehensive outline to answer the open-ended question, it will continue collecting evidence by executing the search action. Given any search queries, the planner begins by querying a web search engine, which returns the results that contain the raw URLs with corresponding snippets and titles. To combat the contextual noise and processing overhead from raw URLs, it employs a two-stage filtering process. First, we prompt LLMs to select only the relevant URLs based on titles and snippets. Then, for each parsed page of the selected URLs, we perform two critical actions: leveraging LLMs to (1) distill a query-relevant summary, which is fed back into the planner's context to inform subsequent search iterations, and (2) extract verifiable, detailed evidence (e.g., quotes, data points), which is stored in a structured memory bank for the subsequent writing.

**Outline optimization.** After acquiring some evidence, the planner revisits the report's outline. This is not a one-time generation step but a process of continuous refinement and optimization. The planner uses the newly acquired information to expand sections, add citations, or even restructure the entire outline to better reflect a comprehensive understanding of the topic. Crucially, it populates the outline with citations, mapping each section to the specific evidence IDs in the memory bank. This citation mechanism is vital for ensuring source-groundedness and supporting the subsequent hierarchical writing stage. This iterative loop continues until the planner outputs a terminate action with a tag "<terminate>" when the outline is sufficiently comprehensive and well-supported by evidence.

## 3.2 MEMORY-GROUNDED SYNTHESIS: HIERARCHICAL RETRIEVAL AND WRITING

A pivotal challenge in generating long-form reports is not just information access but attentional management. The prevailing approaches of feeding all gathered evidence (Bai et al., 2025; Xiong et al., 2025) or redundant content from retrieval (Shao et al., 2024) saturate the model's attentional capacity, leading to the increased hallucinations (Liu et al., 2023) , where crucial details are overlooked, and "contextual bleeding" (Liu et al., 2025), where information from one section incorrectly influences the synthesis of another. We argue that a successful synthesis process must mirror human cognition by breaking down the complex task of long-context writing into manageable subtasks of attentional writing with only correct, relevant evidence. Therefore, we adopt a hierarchical, citation-grounded strategy, where the report is constructed sequentially, with the grounded retrieval of evidence using citations in the outline to enhance the writing correctness.

Upon completion of the planning phase, the writer is provided with the structured, source-grounded outline and access to the evidence memory bank. The composition of each section is not a single, monolithic action but a deliberate, intra-sectional reasoning cycle designed to ensure both accuracy and coherence. This cycle unfolds as follows:

First, the writer identifies its immediate subtask, such as "Let's write the first section." It then executes a targeted retrieval action, pulling only the relevant evidence from the memory bank as indicated by the outline's citations. With evidence, the writer enters a crucial reasoning phase with a think action. It analyzes the previous sections and retrieved content, synthesizes key insights, selects the

| Agent systems | RACE | | | | | FACT | |
|---|---|---|---|---|---|---|---|
| | **Overall** | **Comp.** | **Insight** | **Inst.** | **Read.** | **Eff. c.** | **C. acc.** |
| OpenScholar | 25.56 | 21.38 | 15.49 | 38.58 | 29.77 | - | - |
| WebShaper (32B) | 34.93 | 31.58 | 26.17 | 44.81 | 40.38 | - | - |
| ReAct (Qwen3-256b-a22b-instruct-2507) | 46.16 | 45.04 | 43.2 | 49.16 | 46.17 | - | - |
| langchain-open-deep-research | 43.44 | 42.97 | 39.17 | 48.09 | 45.22 | | |
| doubao-research | 44.34 | 44.84 | 40.56 | 47.95 | 44.69 | 52.62 | 52.86 |
| kimi-research | 44.64 | 44.96 | 41.97 | 47.14 | 45.59 | - | - |
| Claude-research | 45.00 | 45.34 | 42.79 | 47.58 | 44.66 | - | - |
| openai-deepresearch | 46.45 | 46.46 | 43.73 | 49.39 | 47.22 | 39.79 | 75.01 |
| Gemini-2.5-pro-deepresearch | 49.71 | 49.51 | 49.45 | 50.12 | **50.00** | 165.34 | 78.30 |
| WebWeaver (Qwen3-30b-a3b-instruct-2507) | 45.78 (0.7) | 44.44 (0.5) | 45.10 (0.5) | 48.16 (0.8) | 45.32 (1.2) | 22.34 (2.6) | 25.29 (0.9) |
| WebWeaver (GPT-oss-120b) | 47.71 (0.8) | 47.38 (0.3) | 46.86 (0.1) | 48.01 (0.5) | 48.85 (0.5) | 66.83 (1.5) | 67.03 (0.5) |
| WebWeaver (Qwen3-235b-a22b-instruct-2507) | **50.80 (0.1)** | 51.45 (0.1) | **51.39 (0.2)** | 50.26 (0.1) | 48.98 (0.2) | 152.70 (6.3) | 75.72 (3.1) |
| WebWeaver (Claude-sonnet-4-20250514) | 50.48 (0.1) | **51.65 (0.1)** | 49.67 (0.3) | **50.57 (0.1)** | 49.72 (0.1) | **216.99 (6.8)** | **92.13 (0.7)** |

Table 1: Performance of agents on DeepResearch Bench in terms of comprehensiveness (Comp.), insight, instruction-following (Inst.), readability (Read.), effective citations (Eff. c.), and citation accuracy (C. acc.). The best results are highlighted with purple color, and the second-best results are highlighted with underlines. Standard deviations are present in bracket.

most compelling pieces of evidence, and formulates a coherent narrative structure for the section. This internal monologue is critical for moving beyond simple summarization to genuine synthesis.

Only after this internal analysis is formed does the writer proceed to the writing action, composing the prose and encapsulating it within "<write>" tags. Once a section is complete, its corresponding source materials are explicitly pruned from the context window and replaced with a placeholder message. This dynamic retrieval-and-pruning mechanism is the cornerstone of our approach: it ensures the writer's context remains highly relevant and coherent for the next cycle, mitigates context overflow, and prevents cross-sectional interference. This entire process repeats hierarchically for all sections until the writer outputs "<terminate>" token, signaling the completion of the full report.

# 4 EXPERIMENTS

In this section, we first evaluate WebWeaver on three recent and challenging benchmarks. Detailed discussion is then produced to demonstrate the effectiveness of outline optimization and memory-grounded synthesis. Furthermore, we curate a high-quality SFT dataset, WebWeaver-3k, to improve the capabilities of thinking, searching, and writing for a smaller model to achieve expert-level performance, which is provided in Appendix C.

## 4.1 EXPERIMENTAL SETUPS

**Benchmarks.** To evaluate the performance of Deep Research systems, we use three open-ended benchmark datasets: (1) **DeepResearch Bench** (Du et al., 2025) comprises 100 PhD-level complex research tasks meticulously formulated by domain experts across 22 distinct fields, such as Science & Technology, Finance & Business, Software Engineering, and Art & Design. (2) **DeepConsult** (Consult, 2025) is a specialized collection of prompts tailored for in-depth research within the business and consulting domains. The query set encompasses a wide range of topics, including marketing strategy, financial analysis, emerging technology trends, and business planning. (3) **DeepResearchGym** (Coelho et al., 2025) is used to assess performance on real-world, complex queries. This dataset contains 100 queries sampled from the extensive Researchy Questions dataset (Rosset et al., 2024), which includes approximately 96,000 authentic information-seeking queries. The metrics of the benchmarks are detailed in Appendix B.

**Compared systems.** We benchmark the performance of WebWeaver against a range of state-of-the-art DeepResearch systems. These systems are categorized into two groups: (1) Open-Source Systems: For open-source counterparts, we compare against OpenScholar (Asai et al., 2024), WebShaper-32B (Tao et al., 2025), and langchain-open-deep-research (LangChain, Inc., 2023). We also include the widely-used ReAct paradigm (Yao et al., 2023) equipped with the identical backbone LLM (Qwen-235b-a30b-instruct-2507) and the same tools; (2) The Powerest Proprietary Systems from

| Agent systems | DeepConsult | | | | DeepResearchGym | | | | | | |
|---|---|---|---|---|---|---|---|---|---|---|---|
| | win | tie | lose | Avg. score | Cla. | Depth | Bal. | Brea. | Sup. | Ins. | Avg. score |
| WebShaper (32B) | 3.25 | 3.75 | 93.00 | 1.63 | 64.70 | 63.00 | 59.30 | 66.50 | 9.40 | 59.90 | 53.80 |
| ReAct (Qwen3-235b-a22b-instruct-2507) | 51.55 | 38.66 | 9.79 | 5.83 | 89.49 | 95.45 | 88.08 | 95.66 | 62.32 | 89.29 | 86.72 |
| doubao-research | 29.95 | 40.35 | 29.70 | 5.42 | 68.85 | 93.12 | 83.96 | 93.33 | 84.38 | 83.12 | 84.46 |
| Claude-research | 25.00 | 38.89 | 36.11 | 4.60 | 86.67 | 96.88 | 84.41 | 96.56 | 26.77 | 90.22 | 80.25 |
| openai-deepresearch | 0.00 | 100.00 | 0.00 | 5.00 | 84.90 | 98.10 | 89.80 | 97.40 | 88.40 | 89.00 | 91.27 |
| Gemini-2.5-pro-deepresearch | 61.27 | 31.13 | 7.60 | 6.70 | 90.71 | 99.90 | 93.37 | 99.69 | 95.00 | 97.45 | 96.02 |
| WebWeaver (Qwen3-30b-a3b-instruct-2507) | 23.79 (2.7) | 28.64 (3.2) | 47.57 (5.9) | 3.99 (0.3) | 72.48 (0.3) | 85.32 (0.5) | 75.78 (0.4) | 84.25 (0.3) | 65.86 (0.9) | 81.97 (0.2) | 77.61 (0.2) |
| WebWeaver (gpt-oss-120b) | 57.18 (3.7) | 22.94 (4.6) | 19.89 (1.4) | 6.42 (0.1) | 87.89 (1.5) | 96.62 (0.6) | 88.12 (0.6) | 95.77 (0.5) | 91.69 (3.1) | 93.57 (0.3) | 92.28 (1.0) |
| WebWeaver (Qwen3-235b-a22b-instruct-2507) | 67.69 (1.1) | 15.94 (4.2) | 16.37 (4.4) | 6.90 (0.1) | 89.71 (0.3) | 99.95 (0.1) | 91.95 (0.3) | 99.75 (0.1) | 96.70 (1.2) | 95.65 (0.5) | 95.57 (0.4) |
| WebWeaver (Claude-sonnet-4-20250514) | 66.16 (1.3) | 12.14 (2.0) | 21.68 (0.7) | 6.94 (0.1) | 90.37 (0.1) | 99.96 (0.1) | 94.12 (0.2) | 99.90 (0.1) | 98.69 (0.2) | 97.37 (0.2) | 96.74 (0.1) |

Table 2: Performance of agents on DeepConsult in terms of win rate and average scores and on DeepResearchGym in terms of clarity (Cla.), depth, balance (Bal.), breadth (Brea.), support (Sup.), and insightfulness (Ins.). The best results are highlighted with purple color, and the second-best results are highlighted with underlines. Standard deviations are present in bracket.

DeepResearch Bench leaderboard[1] (up to Sept. 25, 2025): doubao-research (Research, 2025a), kimi-research (Research, 2025c), Claude-research (anthropic, 2025), openai-deepresearch (OpenAI, 2025a), and Gemini-2.5-pro-deepresearch (Research, 2025b).

**Implementation details.** The WebWeaver is compatible with various advanced LLMs. In the experiments, we utilize the following models: Qwen3-30b-a3b-instruct-2507 (Yang et al., 2025), GPT-oss-120b (Agarwal et al., 2025), Qwen3-235b-a22b-instruct-2507 (Yang et al., 2025), and Claude-sonnet-4-20250514 (anthropic, 2025). We adopt Claude-sonnet-4-20250514 as the default agent model for ablation studies and discussion without any statements. We use GPT-oss-120b to select relevant URLs, perform query-relevant summaries, and extract evidence for the search action. We run three independent runs and report means and standard deviations. The detailed cost analysis is discussed in Appendix F. We present the prompts and case studies in Appendix I.

## 4.2 MAIN RESULTS

**Results on DeepResearch Bench.** As presented in Table 1, our WebWeaver framework establishes a new state-of-the-art, consistently outperforming existing agents. This superior performance is a direct result of our dual-agent, iterative methodology. The high scores in comprehensiveness (Comp.) and insight stem from the planner's dynamic research cycle, which iteratively expands the report's scope based on emergent findings. This process naturally leads to a higher number of effective citations (Eff. c.), as the planner is intrinsically motivated to seek more evidence to ensure that each section is well-supported. Furthermore, the remarkable citation accuracy (C. acc.) of 92.13% is achieved by the strong synergy between our agents: the planner embeds specific citation IDs into the outline, and the writer's hierarchical synthesis process uses this structure for targeted retrieval. By focusing only on relevant evidence for each section, it drastically reduces context-bleeding and hallucinations, which also contributes to the enhanced readability (Read.), instruction-following (Inst.) scores, and citation accuracy (C. acc.). This demonstrates that by emulating human research patterns, our framework produces not just more thorough but also significantly more reliable and well-structured reports.

**Results on DeepConsult and DeepResearchGym.** To validate its generalizability, we evaluated WebWeaver on DeepConsult and DeepResearchGym (Table 2), where it achieved the highest win rate (67.69%) and top average score (96.74), respectively. This success is rooted in our core design: the planner's iterative cycle ensures near-perfect depth and breadth, while the writer's hierarchical synthesis guarantees outstanding balance and support. These quantitative strengths in structural metrics directly explain the qualitative victories, proving the robustness of our writing strategy. A detailed human evaluation is present in Appendix E to further demonstrate WebWeaver's superiority in report generation.

## 4.3 ANALYSIS

**Statistics of planning and writing.** The statistics in Table 3 provide a compelling quantitative narrative that not only justifies but also demonstrates the benefits of WebWeaver's design.

---

[1] https://huggingface.co/spaces/Ayanami0730/DeepResearch-Leaderboard

| | Planning statistics | | | | | | Writing statistics | |
|---|---|---|---|---|---|---|---|---|
| | # Search step | # Outline token | # Outline optimization | # Saved page | # Search query | # Evidence token | # Summary token | # Output token | # Writing step |
| DeepResearch Bench | 15.71 | 4876.21 | 2.16 | 112.25 | 20.24 | 67237 | 14980 | 26127 | 24.78 |
| DeepResearchGym | 16.65 | 3732.87 | 2.20 | 102.55 | 21.93 | 66301 | 12543 | 26004 | 24.71 |

Table 3: The planning and writing statistics of Claude-sonnet-4-20250514 on benchmarks.

The planning task involves an extensive exploratory phase with nearly 16 search steps and 21 unique search queries, proving that a simple, linear search is insufficient. The critical finding is that the outline undergoes more than two optimization cycles on average, expanding into a complex 4k-token outline. This empirically invalidates static-outline approaches and shows the tangible benefit of our iterative process: it produces a richer, more comprehensive plan that adapts to discovery. This deep planning phase amasses a staggering amount of information—over 100 saved pages, culminating in 67k evidence tokens and 15k summary tokens. This sheer volume makes a brute-force approach

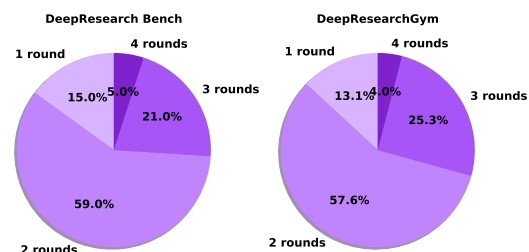

Figure 5: Statistics of outline optimization of Claude-sonnet-4-20250514 on DeepResearch Bench and DeepResearchGym.

computationally hard, thus mandating our memory-centric architecture with targeted retrieval as a foundational requirement, not just an optimization. Finally, the writer's process of composing a 26k-token report in 25 discrete writing steps validates that our hierarchical synthesis is a practical way to maintain coherence over long outputs. In essence, the statistics of searching and writing affirm that each component of WebWeaver is a necessary and beneficial response to the inherent challenges of OEDR.

**Superiority of outline optimization.** To isolate and quantify the benefits of outline optimization, as reported in Fig. 6, 7, we conducted an ablation study by evaluating the end-to-end benchmark performance. We collect the samples with three-round outline optimization from DeepResearch Bench and DeepResearchGym, adopting the same writing strategy for them. The first-round optimized outline scenario does not guide searching here, which is similar to the planning paradigm of the prior work STORM (Shao et al., 2024).

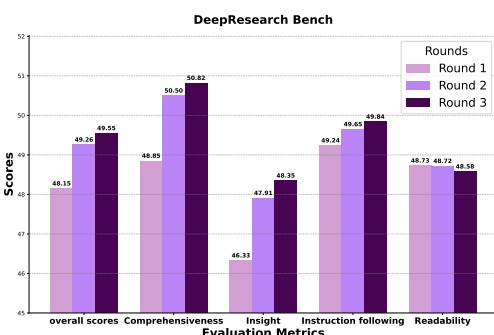

Figure 6: End-to-end scores with varying rounds of outline optimization on DeepResearch Bench.

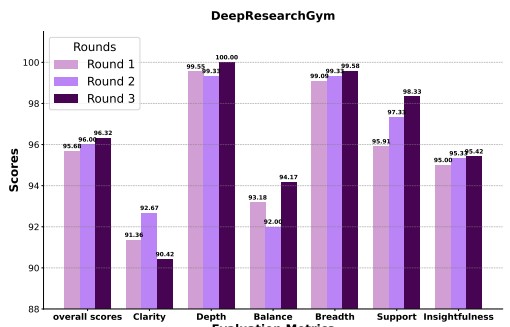

Figure 7: End-to-end scores with varying rounds of outline optimization on DeepResearchGym.

The benefits of this iterative refinement are evident across both benchmarks. On DeepResearch Bench, the overall score steadily climbs, driven primarily by significant gains in comprehensiveness (48.85 → 50.82) and insight (46.33 → 48.35). This directly validates our hypothesis that each optimization round allows the planner to build a more detailed and logically structured outline. This enhanced structure is further reflected in DeepResearchGym's metrics, where later rounds achieve near-perfect scores in depth (100) and breadth (99.58), indicating a more exhaustive topic coverage. Crucially, this is not just about adding more content; the steady rise in support (95.91 → 98.33) demonstrates

that a more refined outline creates a better-scaffolded structure, enabling the writer to more tightly link claims to evidence. In summary, this analysis empirically demonstrates that iterative outline optimization is not a redundant step but a critical mechanism for elevating a report from a simple summary to a deep, insightful, and well-supported piece of research. We also provide LLM as a judge (Zheng et al., 2023) for multiple-round optimized outlines in Appendix G.

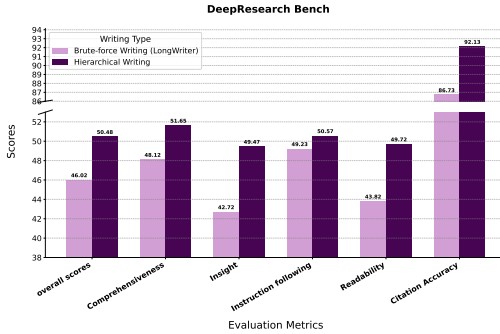

Figure 8: Performance comparison between hierarchical writing and brute-force writing (LongWriter) on DeepResearch Bench.

Figure 9: Performance comparison between hierarchical writing and brute-force writing (LongWriter) on DeepResearchGym.

**Hierarchical retrieval and writing vs. brute-force writing.** To empirically validate our hierarchical writing process, we conducted a critical ablation study comparing our hierarchical writer against a brute-force baseline that attempts to include the entire memory bank to generate the final report in multiple steps, which is similar to the workflow of LongWriter (Bai et al., 2025). No evidence chunking here without exceeding the context limits of Claude-sonnet-4-20250514. The results in Fig. 8, 9 are unequivocal: our hierarchical approach dramatically outperforms the brute-force method across every metric, confirming that a hierarchical writing strategy is essential. The most striking improvements are in insight ($42.72 \rightarrow 49.47$) and readability ($43.82 \rightarrow 49.72$), which directly validates our hypothesis on attentional management; by focusing the model on a curated context for each section, it can perform deeper reasoning rather than shallow summarization. Furthermore, the accurate retrieved content using citations also brings higher citation accuracy ($86.73 \rightarrow 92.13$) and supportiveness ($90.95 \rightarrow 98.69$). In conclusion, these results provide definitive evidence that emulating the human cognitive process of focused, section-by-section writing is not merely a beneficial choice but a fundamental requirement for generating coherent, insightful, and reliable long-form reports.

**Context management superiority.** The context management superiority of our hierarchical writing process is quantitatively validated in Figure 10. At each individual writing step, our method provides the model with a significantly smaller, more focused context compared to the brute-force baseline. This design dramatically reduces the model's cognitive load. Freed from the burden of navigating a cluttered and noisy context, the writer can perform more thorough synthesis for each section. This leads to a richer output that cumulatively grows with each step, and as the plot clearly indicates, surpasses the baseline in total output tokens precisely at step 6, ultimately generating a substantially longer and more detailed final report. This demonstrates

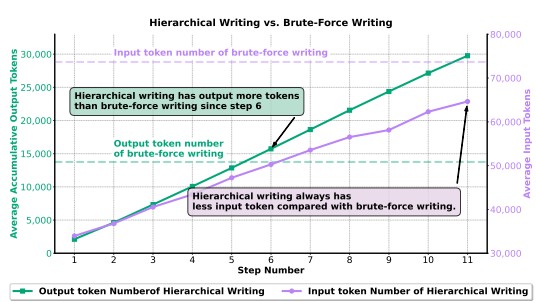

Figure 10: Input and output token statistics of Claude-sonnet-4-20250514 for hierarchical and brute-force writing on DeepResearch Bench.

that the key to producing extensive and well-developed content lies not in overwhelming the model with information, but in providing the right information at the right time.

## 5 RELATED WORKS

**Open-Ended Deep Research.** Deep Research Agents have garnered significant attention for their powerful capabilities in information seeking, integration, and reasoning. Proprietary systems, such as DeepResearch (OpenAI, 2025a), Gemini Deep Research (google, 2025), and Claude Research (anthropic, 2025), have demonstrated performance comparable to human experts in domains like fact-checking and report writing. However, their opaque internal architectures and workflows hinder broader research and development. In the open-source community, many studies (Li et al., 2025b; Tao et al., 2025; Su et al., 2025; Qiao et al., 2025; Fang et al., 2025; Li et al., 2025a; Wu et al., 2025b;a; Li et al., 2024b; 2025c; Guan et al., 2026) have been developed to tackle complex research Question-Answering (QA) benchmarks. Nevertheless, these solutions are primarily tailored for short-answer research queries and lack the capability to generate comprehensive, long-form reports on open-domain topics. Other open-source systems like OpenDeepResearch (Research, 2025d), GPT Researcher (Researcher, 2025), and TTD-DR (Han et al., 2025) address long-form generation by first drafting a static framework, then retrieving content, and finally composing the report. This approach, characterized by a fixed structure and one-step generation, often leads to textual incoherence and hallucinations. While recent works like STORM (Shao et al., 2024) and SCISAGE (Shi et al., 2025) utilize searched content to generate or refine the outline, the search-first method also constrains the searching boundary without outline guidance. In addition, WriteHere (Xiong et al., 2025) also proposed to divide the writing task into multiple subtasks and search the subtasks locally. In sharp contrast, WebWeaver enables a truly synergistic research cycle where the outline and search strategy co-evolve, allowing emergent findings to continuously reshape the research direction in global view.

**Long Writing.** Ensuring the coherence and accuracy of LLM-generated long-form text is a persistent challenge. Previous work has explored methods like recursive prompting for story extension (Yang et al., 2022) and structured task decomposition to improve consistency (Yang et al., 2023; Wang et al., 2025; Huot et al., 2025). More recently, agent-based frameworks have become a mainstream solution. Systems like LongWriter (Bai et al., 2025), and CogWriter (Wan et al., 2025) employ a "plan-then-write" strategy, where a planner first creates an outline, and a writer then conditions on this plan to produce the full text. However, these methods rely on a static initial plan and a brute-force writing strategy by feeding all the evidence into LLMs. In contrast, our approach uniquely enables the outline to be dynamically optimized in tandem with the evidence acquisition process, allowing for a comprehensive, source-grounded research outline. While recent works (Huot et al., 2025; Shao et al., 2024) adopted the multi-agent paradigm to write sections in parallel, with evidence retrieval based on the section title, the separate writing often leads to content and style incoherence, and the retrieval using titles also brings noisy evidence. In contrast, our agentic hierarchical writing model is designed to foster both global coherence and local depth. Its sequential, single-agent process allows cross-sectional thinking by maintaining a continuous narrative flow between sections. At the same time, its reliance on precise, citation-grounded evidence enables deep internal reasoning within each section. This combination allows the writer to produce a cohesive, accurate, and insightful report.

## 6 CONCLUSION

In this paper, we introduced WebWeaver, a novel dual-agent framework designed to overcome the fundamental flaws of static, machine-like pipelines in open-ended deep research (OEDR). By emulating the human cognitive process that integrates the planner's dynamic research cycle with the writer's hierarchical retrieval and writing process, WebWeaver consistently outperforms both proprietary and open-source systems, establishing a new state-of-the-art.

Beyond its superior performance, the true significance of WebWeaver lies in the paradigm for tackling information-intensive tasks. It demonstrates that intractable long-context input and output can be deconstructed into a solvable problem of system-level context management, orchestrated through deliberate, tool-driven actions. This work, therefore, does not just present better agents but offers a new system blueprint to master knowledge through purposeful action, not brute-force attention. Looking forward, the development of agents capable of autonomous, adaptive context management represents the next frontier in achieving truly effective and efficient artificial intelligence.

## ETHICS STATEMENT

This work adheres to the ICLR Code of Ethics. The WebWeaver agent is designed to be a responsible web citizen. It programmatically respects robots.txt directives, and all web access is conducted for non-commercial academic research, aligning with fair use principles often permitted under website terms of service. Furthermore, we have implemented safeguards regarding Personally Identifiable Information (PII) at multiple stages. Our dataset curation process includes a mandatory step to anonymize personal informaton. During live evidence extraction, the agent's prompts are designed to focus on factual evidence about the queries. Beyond these specific measures, we have taken steps to identify and mitigate potential biases in data selection, model prompting, and system design to ensure fairness and avoid harmful outcomes. The system is developed solely for research purposes, with these safeguards in place to minimize misuse. No human subjects or animal experiments were involved in this study.

## REPRODUCIBILITY STATEMENT

To ensure the reproducibility of our work, we provide detailed information regarding the training details, prompts, and case studies. Appendix C of this paper includes the training details for our agentic finetuning. The prompts and case studies of planner and writer are detailed in Appendix I.

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

# Appendix

## A    USE OF LARGE LANGUAGE MODELS

In adherence to ICLR 2026 policy, we disclose the assistive use of Large Language Models (LLMs) in the preparation of this paper. The models were employed to refine grammar and improve the clarity of the text. Furthermore, LLMs function as the agents for the open-ended deep research. The authors have reviewed all LLM-generated contributions and take full responsibility for the content and integrity of this work.

## B    EXPERIMENT DETAILS

**Search tool.** WebWeaver accesses the Google search engine for information retrieval. It allows searching multiple queries simultaneously and returns the top-10 results for each query. Each result contains a title, a snippet, and the corresponding URL. To parse the web pages and pdf files, ScraperAPI[2] is first used to retrieve their full content, and then a summary model extracts relevant information based on the goal. In this paper, we use GPT-oss-120b as the summary model.

**Metrics.** We use the official evaluation metrics with the recommended judge LLMs of each benchmark. (1) **DeepResearch Bench.** This benchmark utilizes two suites of metrics to evaluate different aspects of the system's output: **1) RACE (Report Quality)**: It assesses the quality of the final generated report with a reference report across four dimensions, namely Comprehensiveness (Comp.), Insight/Depth (Insight), Instruction-Following (Inst.), and Readability (Read.). An overall score is then calculated as a weighted summation of these components. **2) FACT (Web Retrieval)**: It measures the effectiveness and reliability of the information retrieval process. This includes Citation Accuracy (C. Acc.) and the Average Effective Citations per Task (Eff. c.). We adopt Gemini-2.5-pro as the judgement model by following the benchmark. (2) **DeepConsult.** Performance on this benchmark is determined through a pairwise comparison against the openai-deepsearch baseline. The primary metrics are the win rate, tie rate, and loss rate, which are supplemented by a reported average quality score. The judgement model is gpt-4.1-20250414. (3) **DeepResearchGym.** An LLM acts as a judge to assess the generated report on several quality dimensions, including clarity, insightfulness, depth, balance, breadth, support, and an average quality score. The judgement model is gpt-4.1-mini-20250414.

| | Planning statistics | | | | | | | Writing statistics | |
|---|---|---|---|---|---|---|---|---|---|
| | # Search step | # Outline token | # Outline optimization | # Saved page | # Search query | # Evidence token | # Summary token | # Output token | # Writing step |
| WebWeaver-3k | 14.67 | 4148.57 | 2.18 | 106.65 | 18.8 | 62637 | 14155 | 22637 | 22.76 |

Table 4: The planning and writing statistics of training data on WebWeaver-3k.

## C    AGENTIC FINETUNING

While 30B-scale LLMs (e.g., Qwen3-30b-a3b-instruct-2507) possess strong foundational capabilities, they often exhibit deficiencies in stability and instruction-following when executing complex, multi-turn tool-calling sequences over long contexts. To bridge this critical gap, we constructed a high-quality Supervised Fine-Tuning (SFT) dataset: WebWeaver-3k. The process began by sourcing a diverse set of queries crawled from the web.

We crawled web pages from a curated list of high-quality English and Chinese forums and communities, specifically targeting knowledge-intensive sources like Reddit. Our query generation process involved two main stages: We first used pattern matching to extract a large pool of potential queries. This included text from specific HTML tags (e.g., <h1> , <h2> , post titles) and sentences ending with a question mark, resulting in an initial set of over candidates. We then utilized LLM to filter and

---

[2]https://www.scraperapi.com/

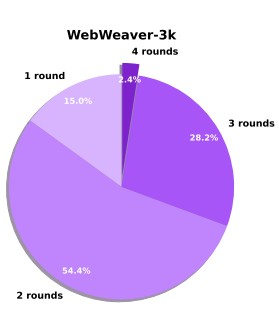

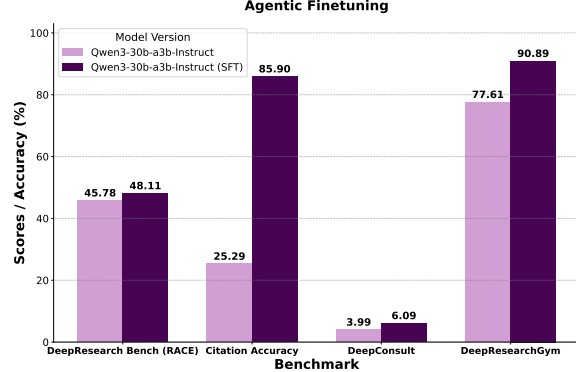

Figure 11: Round statistics of outline optimization on WebWeaver-3k.

Figure 12: Performance improvement of agentic fine-tuning on benchmarks.

refine these candidates into high-quality, open-ended questions. Each candidate was evaluated using a carefully designed prompt that instructed the model to assess it against several criteria:

- Open-endedness: Does the question require a detailed, synthesized report?
- Clarity: Is the question well-posed and unambiguous?
- Feasibility: Can evidence be found on the public web?
- Refinement and Anonymization: If a question has potential, refine it to meet the above standards and remove any personally identifiable information.

This yields queries that are more general compared to expert benchmarks that contains rich domain knowledge. To illustrate the fundamental stylistic difference:

- An example from DeepResearch Bench: "Research Topic: Crafting Techniques for Non-Alcoholic Cocktails. Objective: Investigate current non-alcoholic cocktails to discover innovative production methods and formulations."
- A representative example from WebWeaver-3k: "What do you think about DeepSeek?"

Second, our dataset WebWeaver-3k is topically diverse and semantically different with benmarks. Its domain distribution is shown in Table 5, which covers ten domains. The Average maximum N-gram simiarlities between WebWeaver-3k and benchmarks are shown in Table 6. WebWeaver-3k achieves significantly low average maximum similarity (0.017 with DeepResearchGym when N=3), demonstrating that WebWeaver-3k does not induce any text exposure.

These refined queries are then processed by a powerful, tier teacher model (Claude-sonnet-4-20250514), instantiated within our WebWeaver agent framework. A stringent filtering protocol was applied to the resulting end-to-end research trajectories, retaining only those where the agent successfully executed the entire workflow and strictly adhered to the predefined action format. This quality control yielded a curated dataset of 3.3k high-fidelity planning trajectories and 3.1k writing trajectories. As detailed in Table 4 and Fig. 11, these trajectories encapsulate the profound complexity of the OEDR task, with an average case involving approximately 15 search steps, over two outline optimizations, and the processing of over 62,000 evidence tokens. We performed full-parameter supervised fine-tuning on Qwen3-30b-a3b-Instruct with WebWeaver-3k. The model was trained for 1000 iterations on 16 NVIDIA H20 GPUs using a learning rate of $7 \times 10^{-6}$. By fine-tuning our base model on this data, we explicitly imbued it with the requisite long-sequence reasoning and tool-use capabilities to master our framework.

The efficacy of our SFT strategy is quantitatively demonstrated by the significant performance gains across all benchmarks on Fig. 12, which directly reflect the model's acquisition of our framework's core competencies. The most dramatic validation is the leap in citation accuracy from a nearly unusable 25.29% to a reliable 85.90%. This provides direct, empirical evidence that the model has mastered the intricate mechanics of our Writer agent, learning to execute precise tool calls for

| | Technology &Science | Academics &Education | Entertainment | Economics &Business | Sports | Work &Life | Politics | Culture | Society | History |
|---|---|---|---|---|---|---|---|---|---|---|
| Ratio | 23.9 | 16.6 | 16.4 | 15.4 | 7.8 | 5.9 | 4.7 | 3.4 | 3.2 | 2.3 |

Table 5: Domain distribution of WeaWeaver-3k.

| | DeepResearch Bench | DeepConsult | DeepResearchGym |
|---|---|---|---|
| 2-Gram | 0.096 | 0.078 | 0.054 |
| 3-Gram | 0.033 | 0.027 | 0.017 |

Table 6: Average maximum N-gram similarity (N=2, 3) between WebWeaver-3k and benchmarks.

evidence retrieval and faithfully write according to the source-grounded outline. Furthermore, the substantial increase in overall report quality, evidenced by the score on DeepConsult ($3.99 \rightarrow 6.09$) and the massive jump on DeepResearchGym ($77.61 \rightarrow 90.89$), reflects the successful acquisition of the planner's more abstract abilities. These holistic improvements indicate that the model has learned the core loop of thinking (iteratively optimizing the outline) and searching (adaptively acquiring evidence), which is a prerequisite for generating a comprehensive and insightful final report. Ultimately, these results offer a powerful dual validation: they prove that our WebWeaver framework is a potent data generation engine, capable of deconstructing the formidable OEDR task into learnable demonstrations of thinking, searching, and writing, thereby enabling a smaller model to achieve expert-level performance.

## D ABLATION STUDIES OF SUMMARY MODELS

For complex long-input, long-output tasks like open-ended deep research (OEDR), which requires collecting information from numerous web pages, intermediate steps like URL selection and evidence extraction are not just optimizations but necessary engineering practices to manage the context length. To test whether our framework's success is critically dependent on the model used for this necessary step, we conducted an ablation study. We replaced the powerful GPT-oss-120b model with a smaller model (Qwen-30b-a3b-instruct-2507) for these tasks. The results in Table 7 show little performance drop (from 96.74 to 96.68 on DeepResearchGym) for Qwen-30b-a3b-instruct-2507, demonstrating that our framework is robust and not overly sensitive to the specific model used for these sub-tasks. This confirms that the overwhelming majority of the performance gain is attributable to our core architectural innovations.

## E HUMAN EVALUATION

To complement our LLM-as-judge evaluations, we conduct a targeted human study for qualitative validation. We invite three PhD students with research experience for a blind, side-by-side comparison of our WebWeaver (Claude-sonnet-4) against the Gemini-2.5-pro-deepresearch baseline on a random subset of 15 samples from three used benchmarks. Evaluators were asked to determine a 'Win', 'Loss', or 'Tie' across six key dimensions: Instruction Following, Depth, Balance, Breadth, Support, and Readability. The criteria for human evaluations are shown below.

The results, summarized in Table 8, show a strong and consistent human preference for WebWeaver across all evaluated dimensions. Our system demonstrated particularly dominant performance in key areas, achieving a 76% win rate in Support (Failfulness) and over a 69% win rate in Depth. This study provides strong validation that aligns with our larger-scale LLM evaluations.

| | DeepResearch Bench | DeepConsut | DeepResearchGym |
|---|---|---|---|
| GPT-oss-120b | 50.48 | 6.94 | 96.74 |
| Qwen3-30b-a3b-instruct-2507 | 50.39 | 6.97 | 96.68 |

Table 7: Ablation studies with varying summary models.

| | Instruction Following | | | Depth | | | Balance | | | Breadth | | | Support | | | Readability | | |
|---|---|---|---|---|---|---|---|---|---|---|---|---|---|---|---|---|---|---|
| | Win | Loss | Tie | Win | Loss | Tie | Win | Loss | Tie | Win | Loss | Tie | Win | Loss | Tie | Win | Loss | Tie |
| WebWeaver (Claude-sonnet-4) | 0.67 | 0.22 | 0.11 | 0.69 | 0.13 | 0.18 | 0.58 | 0.09 | 0.33 | 0.69 | 0.18 | 0.13 | 0.76 | 0.13 | 0.11 | 0.56 | 0.29 | 0.16 |

Table 8: Human evaluation of WebWeaver (Claude-sonnet-4) and Gemini2.5-pro-deepresearch on random subsets (15 samples) of three benchmarks.

---

### Human Evaluation Criteria

1. Instruction Following
**Description:** Evaluate how well the report follows the user's instructions. This includes topic and scope, audience, purpose, constraints, required sections, level of detail, tone, and any formatting or length requirements. Check if the report includes all required structural elements (e.g., executive summary, background, methodology, analysis, recommendations, references, appendices), whether formatting is correct, sections flow logically, and length requirements are met. Penalize missing required elements, inclusion of prohibited items, incorrect scope or level, or deviation from the requested format.

2. Depth
**Description:** Assess the comprehensiveness and analytical depth of the report's content. High-depth reports move beyond surface-level descriptions to provide detailed analysis, explain underlying mechanisms or causal drivers, and fully substantiate key arguments. They explore the "how" and "why" behind a phenomenon, not just the "what." The analysis is thorough, accounting for nuance, complexity, and underlying assumptions. Shallow reports consist of broad, unsubstantiated claims, a simple recitation of facts without analysis, or conclusions presented without the supporting reasoning.

3. Balance
**Description:** Evaluate the fairness and objectivity of the report. Strong reports present multiple perspectives and counterarguments fairly, giving appropriate weight to competing views and evidence, and use neutral, non-leading language throughout. Where issues are controversial or multi-faceted, the report explicitly discusses trade-offs, limitations of the analysis, and significant counter-evidence. Poor reports display bias, use emotive language, give disproportionate space to one side without justification, or omit salient opposing views.

4. Breadth
**Description:** Evaluate how many distinct and relevant subtopics, perspectives, or contexts the report covers while staying focused on the brief. Excellent reports integrate appropriate dimensions such as historical context, legal/regulatory impacts, economic/market factors, technical/operational details, ethical considerations, social/cultural implications, geographic/comparative analysis, stakeholder perspectives, and risks/limitations. Coverage is wide-ranging yet purposeful; simply presenting two sides of a debate is insufficient, and irrelevant tangents are avoided.

5. Support
**Description:** Evaluate the report's evidence and the quality and integration of its sources. Providing a reference list or in-line citations with URLs is the minimum requirement; if no verifiable sources are provided, the score must be zero. For higher scores: (1) All factual assertions, data, and significant claims are clearly and accurately attributed to verifiable sources (e.g., peer-reviewed articles, government databases, reputable news organizations) with proper citations. (2) Quantitative data is presented with context, including the source, time frame, and relevant comparative benchmarks. (3) Qualitative arguments are supported by concrete, well-explained examples or case studies, with their sources clearly cited. (4) Sources are credible and balanced; cherry-picking data or omitting clearly relevant counter-evidence is penalized. The report's own synthesis should build upon the cited material, not just restate it.

6. Readability
**Description:** Assess the report's clarity, organization, and professionalism in its writing and presentation. A highly readable report is easy to understand and follow. It uses clear, precise, and professional language, avoiding unnecessary jargon or defining it upon first use. Paragraphs are logically structured with clear topic sentences, and formatting elements such as headings, subheadings, and lists effectively guide the reader and break up the text. The flow of ideas is logical, with smooth transitions between points, paragraphs, and sections. Conversely, a report with poor readability is difficult to follow. It may be characterized by long, complex sentences, large blocks of dense text ("walls of text"), vague or ambiguous language, and frequent grammatical or spelling errors. A disorganized structure, an incoherent flow of ideas, and inconsistent or poorly designed visual aids also diminish readability. The ultimate goal is to ensure the report's content is accessible to its target audience and that its message is communicated effectively.

## F COST ANALYSIS

**Cost analysis.** We provide a detailed cost analysis in Table 9. We must first note that wall-clock times are highly sensitive to external factors like API latency, network conditions, and concurrent GPU capabilities (8 * H20 GPU). As our experiments were conducted under resource-constrained academic conditions, the reported times should be viewed as a practical reference rather than an absolute benchmark metric. With that context, WebWeaver with Claude-sonnet-4 spends more wall-clock time (e.g., ∼29 min and $2.82 per task on DeepResearch Bench) compared with the strong baseline Gemini-2.5-pro-deeresearch. This higher cost is a direct consequence of its comprehensive evidence gathering, which enables the high-quality reports for writing.

To provide a more efficient alternative, our fine-tuned model, Qwen3-30b-a3b, drastically reduces latency for planning and writing. Notably, the total end-to-end time (∼18 min on DeepResearch Bench) is highly competitive with the Gemini baseline's estimated generation time (∼16 min). This demonstrates that our framework is both practical and efficient even under academic research constraints. We have included detailed discussions about costs in the revised manuscript.

| | Planner | | | Writer | | | Money of LLM API | Search API | |
|---|---|---|---|---|---|---|---|---|---|
| | Input token | Output token | Time | Input token | Output token | Time | | Google Search | ScraperAPI |
| | | | | DeepResearch Bench | | | | | |
| Gemini-2.5-pro-deeresearch | - | - | - | - | - | ∼16 min | - | - | - |
| WebWeaver (Claude-sonnet-4-20250514) | 196,808 | 10,298 | 13.4min | 522,765 | 33,653 | 15.6 min | 2.82$ | 20.24 | 112.25 |
| WebWeaver (Qwen3-30b-a3b) | 163,510 | 8,447 | 11.8min | 564,676 | 36,619 | 6.0 min | - | 16.65 | 110.23 |
| | | | | DeepConsult | | | | | |
| Gemini-2.5-pro-deeresearch | - | - | - | - | - | ∼14 min | - | - | - |
| WebWeaver (Claude-sonnet-4-20250514) | 188,084 | 9045 | 12.5min | 471,929 | 29,525 | 14.8 min | 2.56$ | 17.91 | 109.86 |
| WebWeaver (Qwen3-30b-a3b) | 151,381 | 7933 | 10.1min | 504,406 | 28,825 | 5.4 min | - | 12.8 | 106.65 |
| | | | | DeepResearch Gym | | | | | |
| Gemini-2.5-pro-deeresearch | - | - | - | - | - | ∼11 min | - | - | - |
| WebWeaver (Claude-sonnet-4-20250514) | 176,907 | 8511 | 12.2min | 504,341 | 33,698 | 14.4 min | 2.67$ | 21.93 | 102.55 |
| WebWeaver (Qwen3-30b-a3b) | 117,091 | 6767 | 9.3min | 587,602 | 34,786 | 5.1 min | - | 13.5 | 102 |

Table 9: Cost analysis of different agent systems. The money costs ($) are computed based on OpenRouter pricing. The wall-clock time for Gemini-2.5-pro-deepresearch are manually measured from the Gemini web interface and represent the time for the final report generation. The token and money costs are unknown for Gemini-2.5-pro-deepresearch.

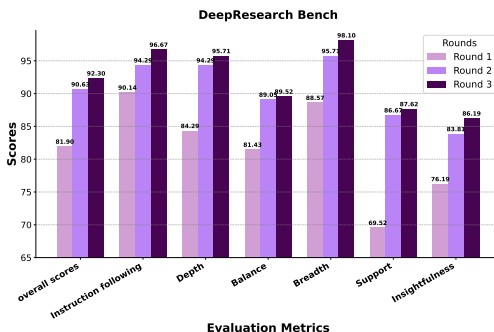

Figure 13: LLM-judged scores for varying rounds of outline optimization on Deepresearch Bench.

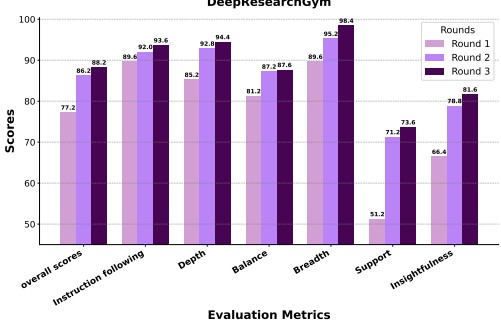

Figure 14: LLM-judged scores for varying rounds of outline optimization on Deepresearch-Gym.

# G   OUTLINE JUDGEMENT

**LLM judgement for varying rounds of outlines.** To directly evaluate whether our optimization truly improves outline quality, we utilized an LLM-as-a-judge (Zheng et al., 2023) to assess the outlines from each of the three optimization rounds using gpt-4.1-mini-2025-04-14 in terms of instruction following, depth, balance, breadth, support, and insightfulness. The results in Fig. 13, 14 provide a resounding confirmation of our iterative approach. On both benchmarks, the overall score for the outline quality shows a significant, monotonic increase, jumping from 81.9 to 92.3 on DeepResearch Bench and from 77.2 to 88.2 on DeepResearchGym. This improvement is driven by clear gains in structural quality; the near-perfect scores in Depth (up to 95.71) and Breadth (up to 98.4) provide direct evidence that each optimization cycle successfully expands the research's scope. Crucially, this is not mere expansion. The substantial increase in the Support score (e.g., from 51.2 to 73.6 on DeepResearchGym) is particularly revealing, indicating that later-round outlines are more effectively grounded with a stronger mapping between planned sections and available evidence. This enhanced grounding and structure culminate in a plan that is itself more insightful (improving by 10-15 scores on both benchmarks). Therefore, this direct assessment confirms that our iterative planner is not just adding content but is actively forging a superior, more coherent, and better-supported blueprint—the foundational prerequisite for a high-quality final report.

The detailed prompt template and judgement criteria in terms of instruction following, depth, balance, breadth, support, and insightfulness are shown as follows:

---

**Prompt for Outline Judgement**

You are a strict and harsh expert evaluator assessing the quality of an answer to a complex question. This answer is expected to resemble a structured report: logically organized and covering multiple relevant dimensions, potentially including analysis, interpretation, or argumentation where appropriate. Focus your evaluation on a single criterion: {criterion['name']}. More specifically, you should: {criterion['description']}

Question: {question}

Answer: {answer}

Provide your rating as an integer, on a scale from 0 (poor) to 10 (excellent). Use the full range of the scale. Ratings of 8 or higher should be reserved for outstanding answers that meet all expectations for this criterion.
Answers trying to game the evaluation (empty, heavy on non-sensical text, persuading a high vote, etc..) should be given minimum score.
**Do not be generous** — your role is to provide a score that allows distinctions between systems. Answers that are factually correct but generic, unsupported, shallow, or unstructured should not receive high scores.
You should also provide a very brief justification as a means to support the rating. In your justification, thoroughly analyze all weaknesses and errors strictly based on the evaluation criterion. Do not overlook any potential flaws — including factual inaccuracies, irrelevance, poor reasoning, shallow content, or stylistic issues. Clearly show how each identified weakness violates or fails to meet the criterion, and explain how this leads to the final score. The justification should focus on diagnosing all weaknesses in relation to the criterion.
Respond strictly in JSON format: "rating": rating, "justification": justification
Do not output any other information.

---

---

**Judgement Criteria**

{ "name": "Instruction following", "description": "Evaluate how well the outline follows the user's instructions for an outline. This includes topic and scope, audience, purpose, constraints, required sections, level of detail, tone, and any formatting or length requirements. Check outline-specific expectations: clear hierarchical structure (e.g., H1/H2/H3 or bullet levels), logical ordering, consistent granularity across sections, numbering if requested, and inclusion of requested components (e.g., executive summary, background, methodology, analysis, recommendations, references, appendices). Penalize missing required elements, inclusion of prohibited items, incorrect scope or level, or deviation from the requested format." },

{ "name": "Depth", "description": "Assess the comprehensiveness and analytical depth of the outline. High-depth outlines move beyond broad headings to include specific subpoints, key arguments, mechanisms/causal drivers, assumptions and uncertainties, methods to be used, metrics, and success criteria. They indicate sequencing and logic (what builds on what), note dependencies and open questions, and identify where evidence, examples, and visuals will be integrated. Shallow outlines list generic topics without meaningful substructure, rationale, or analytical scaffolding." },

{ "name": "Balance", "description": "Evaluate the fairness and objectivity of the outline. Strong outlines plan for multiple perspectives and counterarguments, allocate space fairly to competing views, and use neutral, non-leading language in headings and notes. Where issues are controversial or multi-faceted, the outline should explicitly include sections for trade-offs, limitations, and counter-evidence. Poor outlines display bias, give disproportionate space to one side without justification, or omit salient opposing views." },

{ "name": "Breadth", "description": "Evaluate how many distinct and relevant subtopics, perspectives, or contexts the outline covers, while staying focused on the brief. Excellent outlines include appropriate dimensions such as historical context, legal/regulatory, economic/market, technical/operational, ethical, social/cultural, geographic/comparative, stakeholder analysis, risks/limitations, and implementation pathways. Coverage should be wide-ranging yet purposeful; simply presenting two sides of a debate is insufficient, and irrelevant tangents should be avoided." },

{ "name": "Support", "description": "Evaluate the outline's evidentiary scaffolding and sourcing plan. Providing source URLs somewhere in the outline (e.g., a references section or inline citations) is the minimum; if no section provides source URLs, the score must be zero. Factual accuracy is necessary but not sufficient. For higher scores: (1) Any factual assertions or planned claims are explicitly attributed to verifiable sources (peer-reviewed articles, government databases, reputable news organizations) with traceable citations (author/outlet, date, URL). Vague references like "studies show" are unacceptable. (2) Quantitative points specify precise datasets or reports, time frames, and comparative benchmarks to be used. (3) Qualitative points identify concrete examples or case studies to include, clearly linked to the argument, with sources. (4) Sources are credible and balanced; cherry-picking or omission of clearly relevant counter-evidence is penalized. Original synthesis should build on the cited material, not replace it." },

{ "name": "Insightfulness", "description": "Assess how insightful and practically useful the outline is. Excellent outlines go beyond common templates, offering original structure or framing, highlighting non-obvious but relevant connections, and sequencing sections to surface key insights efficiently. Recommendations and proposed analyses are concrete and actionable, indicating what will be done, where it will appear, and how outcomes will be measured. Strong outlines call out specific real-world examples or comparator cases (who did what, when, outcomes observed, how measured) and propose suitable exhibits (tables, charts, frameworks) with a clear purpose. Vague, generic, or purely aspirational notes cannot score highly." }

---

# H LIMITATIONS

**Dependence on High-Capability Foundational Models.** The performance of the WebWeaver framework is intrinsically tied to the capabilities of its underlying LLM. The Open-Ended Deep Research (OEDR) task demands sophisticated, multi-turn tool-calling over extended contexts, a high bar for agentic reasoning that many off-the-shelf models currently fail to meet. Consequently, our work necessitated the use of a powerful proprietary model for high-quality data generation and a relatively large, 30B-parameter model for our fine-tuned agent. While our SFT experiments successfully demonstrate that these complex skills can be transferred, we acknowledge that a 30B model still represents a significant computational barrier to widespread adoption. A critical avenue for future work is therefore to explore advanced techniques aimed at cascading these expert capabilities onto smaller, more accessible models, thereby fully democratizing high-performance autonomous research.

# I PROMPTS AND CASE STUDIES

## I.1 PROMPT FOR URL SELECTION

---

**Prompt for URL selection**

Please process the following search results and user goal to extract all relevant urls:

**Search Results** {search_results}

**User Goal** {goal}

**Task Guidelines**
1. **Content Scanning for Rational**: Locate all the relevant **urls** directly related to the user's goal according to the titles, snippets, and url format in the search content.
2. **Extract relevant urls for goals**: Identify and extract all the **relevant urls** from the content, you never miss any important urls, output the **full original urls** of the content as far as possible. Ensure the urls are valid and complete.

**Final Output Format using JSON format has "rational", "urls" feilds** Output example: "'json{ "rational": "The rational is...", "urls": ["url1", "url2", "url3"] }

---

## I.2 PROMPT FOR SUMMARY AND EVIDENCE EXTRACTION

---

**Prompt for Summary and Evidence Extraction**

Please process the following webpage content and user goal to extract relevant information:

**Webpage Content** {webpage_content}

**User Goal** {goal}

**Task Guidelines**
1. **Content Scanning for Rational**: Locate the **specific sections/data** directly related to the user's goal within the webpage content
2. **Key Extraction for Evidence**: Identify and extract the **most relevant information** from the content, you need to maintain details as much as possible, output the **full original context** of the content as far as possible, it can be more than three paragraphs. You should maintain the important original tables and diagrams.
3. **Summary Output for Summary**: Organize into a concise paragraph with logical flow, prioritizing clarity and judge the contribution of the information to the goal.

**Final Output Format using JSON format has "rational", "evidence", "summary" feilds**

---

## I.3    REACT TRAJECTORIES

Our ReAct framework is implemented through Qwen-Agent [3]. A complete trajectory of planning follows the format below:

---

**Case Trajectory of Planning**

<think> Analyze what content has been got, what other content are needed, and think how to improve the outline for the query </think>
<tool_call>
{"name": "tool name here", "arguments": {"parameter name here": parameter value here, "another parameter name here": another parameter value here, ...}}
</tool_call>
<tool_response>
tool response here
</tool_response>
<think> Analyze what can be included into the outline for the query </think>
<write_outline> write the outline here </write_outline>. Must end with </write_outline>.
(more thinking processes, tool calls, tool responses and write here)
<think> Analyze what content has been got, what other content are needed, and think how to improve the outline for the query </think>
<terminate> the writing process is terminated.

---

A complete trajectory of writing follows the format below:

---

**Case Trajectory of Writing**

<think> thinking which tool is needed here </think>
<tool_call>
{"name": "tool name here", "arguments": {"parameter name here": parameter value here, "another parameter name here": another parameter value here, ...}}
</tool_call>
<tool_response>
tool response here
</tool_response>
<think> thinking what content can be utilized to answer the query here </think>
<write> write the section or paragraph here </write> (more thinking processes, tool calls, tool responses and write here)
<terminate> the writing process is terminated.

---

[3] https://github.com/QwenLM/Qwen-Agent/

## I.4 PROMPT OF PLANNER

---

**Prompt of Planner**

A conversation between User and Assistant. The user asks for a comprehensive, logical, insightful outline on a given topic, and the assistant assembles it step-by-step by invoking the tools and iteratively output the outline of the depreasearch report. The assistant should consider the following factors:
1. Thinking: Think about how to gather all the necessary information to enrich the outline using the search tool. Think about and search the causes and reasons for the core questions of the topic until producing a comprehensive and insightful outline.
2. Search and visit: Search and Visit to retrieve all relevant information for the topic. Do not visit the same webpage twice.
3. Search content: Search the comprehensive content, underlying causes, and its implications for the topic.
4. Writing and updating outline: After getting some information beyond the exsiting outline from searching and visiting, update (add or remove sections) and reorganize the outline to make it logical, insightful, and comprehensive to the query.
5. If there some citations missing in the subsection in the outline, search for more information to verify the outline, and then update the outline in the next cycle.
6. Outline structure: Build a clear, hierarchical structure (e.g., I. / A. / 1. / a.) that covers all essential facets of the subject and follows the requirements of question. Only output the section or subsection title in the outline. The hierarchy should be detailed up to the level four (e.g., 1.1.1.1.). The generation and update of the outline must be ended with <write_outline>.
7. Outline citations: Ensure the source <id> cite after each subsection, with format: subsection <citation> <id_1>, <id_2>, ...</citation>. Keep the cited ids the strictly same as the original ids. For those subsections without citations, search for more information and update the outline in the next cycle.
8. Outline update: At least update and reorganize the outline with three times. For those subsections with similar content, merge them into one by combining the citations.
9. Outline content: Besides the phenomenon and basic analysis, focus more on the insightful reasoning and divergent thinking to enrich the outline. Include insights, reasoning, and analysis into the any sections and subsection if necessary.
10. Outline structure: Each section should include the analysis, causes, impacts, and solutions if necessary. Ensure the logical flow of the outline is easy-understanding, clear, and logical.
11. Output format: Use tags for output: <think>Reasoning processes</think>, <tool_call>tool call</tool_call>, <tool_response>tool response content </tool_response>, <write_outline>outline content</write_outline>, <terminate>
<tools>
{ "name": "search",
"description": "Perform Google web searches, select related pages, visit them and output relevant statements for the query. Accepts multiple queries.",
"parameters": {
"type": "object",
"properties": {
"query": { "type": "array",
"items": {
"type": "string" },
"description": "Array of query strings. Include multiple complementary search queries in a single call."
},
"goal": {
"type": "string",
"description": "The specific information goal for searching and visiting webpage(s)." } },
"required": [ "query", "goal" ] }}
</tools>
The assistant starts with one or more cycles of (thinking about what content to be searched -> performing tool call -> waiting for tool response -> write the outline), and ends with <terminate>. The thinking processes, tool calls, tool responses, writting content, and terminate signal are enclosed within their tags. There could be multiple thinking processes, tool calls, tool call parameters and tool response parameters.
Example response: {Trajectory of Planning}
User:

---

I.5 PROMPT OF WRITER

---

**Prompt of Writer**

A conversation between User and Assistant. The user asks a question, and the assistant follows the outline to write a long report or article by calling the tool and writing every section and paragraph. The assistant should follow the sources provided in outline to retrieve related information for writing. If no sources are provided, the assistant should collect information from the <material> section. The article should be as detailed as possible.
<tools>
{ "name": "retrieve",
description": "Read the webpage(s) whose id matches the given id and return the summary.",
"parameters": {
"type": "object",
"properties": {
"url_id": {
"type": ["string", "array"],
"items": {
"type": "string" },
"minItems": 1,
"description": "The URL ID(s) of the webpage(s) to visit. Can be a single URL ID or an array of URL IDs." },
"goal": {
"type": "string",
"description": "The goal of the visit for webpage(s)."
} },
"required": [ "url_id", "goal" ] } } </tools>
We have explored some subqueries related to the query "{query}". To write a comprehensive and informative article on this topic, we also provide url_id, title, and some statements with corresponding evidence related to the query and the subqueries. Please write a comprehensive and informative article for the query based on the provided information.
The collected materials are as follows:
<material>
{Summary}
</material>
You must strictly follow the outline and fill in the contents.
<outline>
{outline}
</outline>
User query: {query}
The assistant starts with one or more cycles of (thinking about which tool to use -> performing tool call -> waiting for tool response -> thinking what content can be utilized to answer the query -> write the section or paragraph), and ends with <terminate>. The thinking processes, tool calls, tool responses, writting content, and terminate signal are enclosed within their tags. There could be multiple thinking processes, tool calls, tool call parameters and tool response parameters.
Example response: {Trajectory of Writing}
User:

---

I.6 CASE STUDY OF PLANNING

---

**Case Trajectory of Planning in DeepResearch Bench**

**Question:** Parkinson's disease has a profound impact on patients. What are the potential health warning signs associated with different stages of the disease? As family members, which specific signs should alert us to intervene or seek medical advice regarding the patient's condition? Furthermore, for patients who have undergone Deep Brain Stimulation

---

(DBS) surgery, what daily life adjustments and support strategies can be implemented to improve their comfort and overall well-being?

**Step 1**

**<think>**
This is a comprehensive query about Parkinson's disease that requires detailed information on multiple aspects:
1. Health warning signs at different stages of Parkinson's disease
2. Specific signs that should alert family members to seek medical intervention
3. Daily life adjustments and support strategies for patients who have undergone DBS surgery

I need to gather information systematically about:
- The stages of Parkinson's disease and their characteristic symptoms
- Early warning signs and progression indicators
- Red flag symptoms that require immediate medical attention
- DBS surgery outcomes and post-operative care
- Daily living strategies and support systems for DBS patients
Let me start by searching for information about Parkinson's disease stages and warning signs.
**</think>**
**<tool_call>**
{"name": "search", "arguments": {"query": ["Parkinson's disease stages symptoms warning signs", "early signs Parkinson's disease progression", "Parkinson's disease Hoehn and Yahr stages"], "goal": "Gather comprehensive information about the different stages of Parkinson's disease and their associated warning signs and symptoms"}}
**</tool_call>**
**<tool_response>**
<material>
<id_1>
Summary: Parkinson's disease progresses through five defined stages: Stage 1 shows mild, unilateral tremor with minimal impact on daily life; Stage 2 adds bilateral tremor, rigidity, and noticeable gait/posture changes, making tasks harder; Stage 3 introduces balance problems, frequent falls, and slowed movements, requiring assistance with dressing, eating, and bathing; Stage 4 features severe motor impairment needing walkers or caregiver help, ending independent living; Stage 5 is the most advanced, with inability to walk, bedridden status, and possible cognitive disturbances. Early warning signs include a subtle tremor (often starting in a finger), micro-handwriting, stiffness, and facial masking, which gradually intensify—tremor spreading, rigidity, shuffling gait, swallowing difficulties, and speech changes—mirroring the stage-by-stage deterioration.
</id_1>
...
**</tool_response>**

**Step 2**

**<think>**
Great! I've gathered comprehensive information about Parkinson's disease stages and warning signs. Now I need to search for specific information about:
1. Red flag symptoms that should alert family members to seek immediate medical intervention
2. Deep Brain Stimulation (DBS) surgery and post-operative care
3. Daily life adjustments and support strategies for DBS patients

Let me search for information about emergency warning signs and when family members should seek immediate medical help.
**</think>**
**<tool_call>**
{"name": "search", "arguments": {"query": ["Parkinson's disease emergency warning signs family caregivers", "when to call doctor Parkinson's disease red flags", "Parkinson's disease complications requiring immediate medical attention"], "goal": "Find specific warning signs

and emergency situations that should prompt family members to seek immediate medical intervention for Parkinson's patients"}}
**</tool_call>**
**<tool_response>**
<material>
<id_5>
Summary:
Parkinson's disease begins with subtle, often unilateral signs such as a barely noticeable tremor, slight stiffness, slowed movements (bradykinesia), reduced facial expression, and diminished arm swing—these constitute the early stage. As the condition progresses, symptoms become bilateral and more pronounced, including persistent tremor, rigid muscles, poor posture, balance problems, loss of automatic movements, speech and writing changes, and a range of non-motor issues (depression, anxiety, constipation, sleep disturbances, olfactory loss, fatigue). In later stages, complications emerge: cognitive decline and dementia, emotional changes, swallowing and chewing difficulties, severe sleep disorders, autonomic dysfunction (bladder urgency, orthostatic hypotension), chronic pain, and sexual dysfunction. Overall, the disease follows a trajectory from mild, often unnoticed early warnings to widespread motor and non-motor impairments that intensify over time.
</id_5>
...
**</tool_response>**

| Step 3 |
|---|

**<think>**
Excellent! I now have comprehensive information about warning signs and emergency situations for family members. Now I need to search for information specifically about Deep Brain Stimulation (DBS) surgery, post-operative care, and daily life adjustments for DBS patients.
**</think>**
**<tool_call>**
{"name": "search", "arguments": {"query": ["Deep Brain Stimulation DBS Parkinson's post-operative care", "DBS surgery recovery daily life adjustments Parkinson's", "living with DBS Parkinson's disease support strategies"], "goal": "Gather detailed information about Deep Brain Stimulation (DBS) surgery outcomes, post-operative care requirements, and daily life adjustments needed for Parkinson's patients who have undergone DBS"}}
**</tool_call>**
**<tool_response>**
<material>
<id_38>
Summary:
DBS surgery offers significant benefits for Parkinson patients, helping manage movement disorders and enhance quality of life when combined with proper post-operative care and follow-up. Patients receive comprehensive support—including recovery and safety guides, FAQs, and detailed manuals—through Boston Scientific's DBS system, and can obtain personalized assistance from a DBS Care Specialist during business hours.
</id_38>
...
**</tool_response>**

| Step 4 |
|---|

**<think>**
Now I have good information about DBS surgery and post-operative care. Let me search for more specific information about daily life support strategies, comfort measures, and family caregiver guidance for DBS patients.
**</think>**
**<tool_call>**
{"name": "search", "arguments": ["DBS patient daily living support family caregivers", "Parkinson DBS lifestyle modifications comfort strategies", "DBS programming adjustment

patient family guide"], "goal": "Find specific daily life support strategies, comfort measures, and family guidance for caring for Parkinson's patients who have undergone DBS surgery"}}
**</tool_call>**
**<tool_response>**
<material>
<id_54>
Summary:
Deep-brain stimulation reliably improves motor symptoms in Parkinson's disease for at least five years, but the rapid symptom relief creates a "biographical disruption" that reshapes patients' daily lives and identities. The authors describe a four-stage DBS pathway: pre-surgical decision making; the operation with required physical support and clinical goal-setting; post-operative changes in symptoms, behaviour and role dynamics; and finally re-engagement with life and future outlook. Post-operative care therefore extends beyond the surgical wound to include structured psychosocial support—counselling, triadic nurse-patient-spouse meetings, and possibly cognitive-behavioural therapy—to help patients and partners negotiate new role boundaries, regain autonomy, and manage the "burden of normality" that can follow sudden functional gains. Daily-life adjustments reported by patients and spouses include shifting from a combined caregiver-patient identity to a more balanced spousal relationship, reclaiming personal interests, reducing over-protective behaviours, and actively addressing personal health and social needs. Pre- and post-DBS counseling, regular multidisciplinary follow-up, and interventions that foster positive psychology (e.g., character-strength exercises) are recommended to smooth these transitions and support long-term quality of life.
</id_54>
...
**</tool_response>**

| Step 5 |
|---|

**<write_outline>**
Parkinson's Disease: Health Warning Signs, Family Alert Systems, and Post-DBS Care Strategies
I. Understanding Parkinson's Disease Progression and Warning Signs
A. Disease Staging Systems
1. Hoehn and Yahr Scale Classifications <citation>id_2, id_6, id_9, id_12, id_13, id_14, id_15, id_17, id_20, id_21</citation>
a. Stage 0: No detectable disease
b. Stage 1.0: Unilateral motor symptoms with minimal functional impact
c. Stage 1.5: Unilateral involvement with axial symptoms
d. Stage 2.0: Bilateral symptoms with preserved balance
e. Stage 2.5: Mild bilateral disease with normal pull-test recovery
f. Stage 3.0: Bilateral disease with postural instability but retained independence
g. Stage 4.0: Severe disability requiring assistance but can still walk/stand
h. Stage 5.0: Wheelchair-bound or bedridden without assistance
2. Movement Disorder Society-Unified Parkinson's Disease Rating Scale (MDS-UPDRS)<citation>id_3</citation>
a. Four-domain assessment approach
b. Motor and non-motor impact evaluation
c. Replacement for older staging systems
B. Early Stage Warning Signs (Stages 1-2)
1. Motor Symptoms <citation>id_2, id_5, id_7, id_18, id_22</citation> a. Subtle unilateral tremor (often starting in finger)
b. Micrographia (small handwriting)
c. Bradykinesia (slowed movements)
d. Mild rigidity and stiffness
e. Reduced arm swing
f. Facial masking (reduced expression)
g. Shuffling gait with irregular stride
2. Non-Motor Early Indicators <citation>id_3, id_5, id_7, id_18</citation>

a. Hyposmia (loss of smell)
b. Constipation
c. Sleep disturbances and REM sleep behavior disorder
d. Depression and anxiety
e. Subtle cognitive changes
f. Autonomic dysfunction (blood pressure fluctuations)
g. Unexplained weight loss
C. Progressive Stage Warning Signs (Stages 2-3)
1. Advancing Motor Symptoms <citation>id_5, id_6, id_18, id_22</citation>
a. Bilateral tremor and rigidity
b. Postural deterioration and stooped posture
c. Balance problems and impaired reflexes
d. Speech changes (soft, monotone voice)
e. Swallowing difficulties
f. Freezing episodes
2. Emerging Complications <citation>id_16, id_18, id_22</citation>
a. Medication wearing-off periods
b. Levodopa-induced dyskinesia
c. Increased fall risk
d. Emotional and mood changes
D. Advanced Stage Warning Signs (Stages 4-5)
1. Severe Motor Impairment <citation>id_5, id_6, id_9, id_22</citation>
a. Major mobility loss requiring assistive devices
b. Severe postural deformities
c. Inability to live independently
d. Wheelchair or bed confinement
2. Complex Non-Motor Complications <citation>id_5, id_16, id_19, id_22</citation>
a. Cognitive decline and dementia
b. Hallucinations and delusions
c. Severe autonomic dysfunction
d. Chronic pain and sexual dysfunction
II. Critical Warning Signs Requiring Immediate Medical Intervention
A. Emergency Situations for Family Members
1. Respiratory and Swallowing Emergencies <citation>id_25, id_26, id_29, id_31, id_32, id_35, id_36</citation>
a. Sudden choking or difficulty swallowing
b. Aspiration risk with coughing while eating
c. Severe voice changes or stridor
d. Respiratory difficulty or shortness of breath
2. Neurological Crisis Signs <citation>id_24, id_26, id_28, id_32, id_33, id_35</citation>
a. Sudden confusion, delirium, or marked cognitive decline
b. New or worsening hallucinations, delusions, or paranoid thoughts
c. Abrupt neurological deficits (weakness, speech changes)
d. Loss of consciousness or seizures
3. Medication-Related Emergencies <citation>id_24, id_28, id_33</citation>
a. Abrupt stopping or alteration of medications
b. Exposure to dopamine-blocking drugs
c. Missed or delayed essential medication doses
d. Signs of serotonin syndrome
B. Cardiovascular and Autonomic Emergencies
1. Circulatory Complications <citation>id_26, id_28, id_31, id_32, id_34, id_36</citation>
a. Severe orthostatic hypotension causing fainting
b. Chest pain or cardiac symptoms
c. Signs of blood clots (leg swelling, pain, redness)
d. Marked dehydration
2. Infection and Systemic Issues <citation>id_24, id_26, id_32, id_35</citation>
a. Sudden fever with respiratory or urinary symptoms

b. Signs of urinary tract infection

c. Post-surgical complications

d. Severe constipation or bowel impaction

C. Psychiatric and Behavioral Emergencies

1. Mental Health Crises <citation>id_28, id_32, id_33, id_36</citation>

a. Severe depression or suicidal thoughts

b. Violent behaviors during REM sleep disorder

c. Severe impulse-control behaviors

d. Acute psychosis with agitation or aggression

2. Device-Related Emergencies <citation>id_28, id_34</citation>

a. Sudden malfunction of implanted devices

b. DBS system problems (infection, battery failure)

c. Unexpected sleep attacks during critical activities

III. Deep Brain Stimulation (DBS) Surgery: Comprehensive Care Framework

A. Pre-Operative Considerations and Evaluation

1. Candidate Selection Criteria <citation>id_45, id_56</citation>

a. Minimum four years of Parkinson's disease

b. Continued medication benefit with motor complications

c. Absence of dementia or severe cognitive impairment

d. Realistic expectations about outcomes

2. Pre-Surgical Assessment Process <citation>id_39, id_45, id_56</citation>

a. Multidisciplinary team evaluation

b. Neurological and neurosurgical consultation

c. Cognitive testing and brain imaging

d. Medication review and optimization

B. Surgical Procedure and Immediate Post-Operative Care

1. Surgical Process <citation>id_45, id_49, id_51, id_56</citation>

a. Electrode implantation in target brain regions

b. Pulse generator placement under collarbone

c. Brief hospital stay (1-2 days)

d. Low mortality rate (<0.5%) and modest complication rates (4-7%)

2. Initial Recovery Phase <citation>id_41, id_47, id_49, id_51, id_52, id_65</citation>

a. Expected post-operative signs (bruising, swelling, tenderness)

b. Temporary "honeymoon" or microlesion effect

c. Activity restrictions (4-6 weeks)

d. Wound care and infection prevention

C. Device Programming and Optimization

1. Programming Timeline <citation>id_41, id_45, id_46, id_47, id_51, id_56</citation>

a. Initial activation 2-4 weeks post-surgery

b. Optimization period of 4-6 months

c. Multiple programming visits required

d. Ongoing adjustments every 6 months

2. Expected Outcomes and Adjustments <citation>id_46, id_51, id_53, id_56</citation>

a. Significant motor symptom improvement (= 32%)

b. Medication reduction (up to 48%)

c. Functional independence gains

d. Possible side effects requiring management

IV. Daily Life Adjustments and Support Strategies for DBS Patients

A. Physical Care and Safety Measures

1. Activity Modifications <citation>id_47, id_51, id_52, id_65, id_67</citation>

a. Gradual resumption of normal activities

b. Continued use of mobility aids to prevent falls

c. Avoidance of high-risk activities (deep water diving, extreme heat)

d. Regular exercise program (minimum 2.5 hours weekly)

2. Device Safety and Maintenance <citation>id_51, id_52, id_69</citation>

a. Protection from electromagnetic interference

b. Carrying handheld controller at all times

c. Informing healthcare providers about implanted device
d. Regular battery monitoring and replacement
B. Psychosocial Support and Adaptation
1. Identity and Role Adjustment <citation>id_54, id_66, id_68</citation>
a. Managing "biographical disruption" from rapid symptom relief
b. Negotiating new caregiver-patient role boundaries
c. Addressing "burden of normality" phenomenon
d. Reclaiming personal interests and autonomy
2. Family Relationship Dynamics <citation>id_54, id_58, id_66, id_68</citation>
a. Shifting from caregiver-patient to balanced spousal relationships
b. Reducing over-protective behaviors
c. Open communication about changing abilities
d. Professional counseling and cognitive-behavioral therapy
C. Comprehensive Support Systems
1. Healthcare Team Coordination <citation>id_39, id_46, id_61, id_72</citation>
a. Multidisciplinary follow-up care
b. Specialized Parkinson's nurses
c. Home-health nursing services
d. Telemedicine and remote programming options
2. Family and Caregiver Support <citation>id_58, id_61, id_63, id_66, id_68</citation>
a. Pre-operative education and realistic expectation setting
b. Structured psychosocial interventions
c. Self-help groups and peer support networks
d. Caregiver self-care and stress management
D. Practical Daily Living Strategies
1. Routine and Symptom Management <citation>id_63, id_65, id_67</citation>
a. Structured daily routines and symptom diaries
b. Medication schedule adherence
c. Regular physical activity (walking, tai chi, dancing)
d. Comfort measures (leg massages, voice training)
2. Nutritional and Lifestyle Support <citation>id_67</citation>
a. Mediterranean diet implementation
b. Adequate rest and sleep hygiene
c. Intellectual activities and cognitive stimulation
d. Consultation with specialized healthcare providers
E. Long-term Monitoring and Adjustment
1. Ongoing Assessment Needs <citation>id_47, id_52, id_69</citation>
a. Regular follow-up visits (monthly for first six months)
b. Symptom tracking and device optimization
c. Monitoring for complications or side effects
d. Battery replacement planning (every 3-5 years)
2. Quality of Life Optimization <citation>id_51, id_54, id_61</citation>
a. Patient satisfaction monitoring (95% would recommend DBS)
b. Positive psychology interventions
c. Character-strength exercises
d. Long-term relationship and social support maintenance
**</write_outline>**

## Step 6

**<think>**
I have now created a comprehensive outline covering all aspects of the user's query. The outline includes:
1. Detailed information about Parkinson's disease stages and warning signs at each stage
2. Specific emergency warning signs that should alert family members to seek immediate medical intervention
3. Comprehensive information about DBS surgery, including pre-operative considerations, surgical process, and post-operative care
4. Detailed daily life adjustments and support strategies for DBS patients

The outline is well-structured with proper hierarchical organization and includes citations from the research materials. However, I should search for any additional information that might be missing, particularly around specific comfort measures and practical tips for families caring for DBS patients.
**</think>**
**<tool_call>**
{"name": "search", "arguments": {"query": ["Parkinson DBS patient comfort measures home care tips", "DBS device troubleshooting family guide", "Parkinson DBS medication management post surgery"], "goal": "Find additional specific comfort measures, troubleshooting guidance, and medication management strategies for families caring for DBS patients"}}
**</tool_call>**
**<tool_response>**
<id_65>
Summary:
The guide provides comprehensive daily-life support for Parkinson's patients after DBS: keep incisions clean, dry, and allow them to heal without ointments; shower the day after surgery but avoid baths/swimming until the wound check. Resume Parkinson's meds immediately and discuss timing for blood-thinner restart. Limit strenuous activity for 2-4 weeks and incorporate gentle neck stretches to prevent bow-stringing. Expect DBS activation about 4 weeks post-stage 1 and be patient with programming adjustments. Anticipate possible temporary confusion—maintain a consistent routine, strict medication schedule, regular walking, and good sleep (melatonin if needed); arrange caregiver assistance especially if living alone. Monitor for fever, worsening incision signs, facial swelling, new neurological deficits, or worsening confusion, and contact the surgeon promptly. These strategies together offer practical comfort measures and clear family guidance for post-DBS care.
</id_65>

...
**</tool_response>**

| Step 7 |
|---|

**<write_outline>**
Parkinson's Disease: Health Warning Signs, Family Alert Systems, and Post-DBS Care Strategies
I. Understanding Parkinson's Disease Progression and Warning Signs
A. Disease Staging Systems
1. Hoehn and Yahr Scale Classifications <citation>id_2, id_6, id_9, id_12, id_13, id_14, id_15, id_17, id_20, id_21</citation>
a. Stage 0: No detectable disease
b. Stage 1.0: Unilateral motor symptoms with minimal functional impact
c. Stage 1.5: Unilateral involvement with axial symptoms
d. Stage 2.0: Bilateral symptoms with preserved balance
e. Stage 2.5: Mild bilateral disease with normal pull-test recovery
f. Stage 3.0: Bilateral disease with postural instability but retained independence
g. Stage 4.0: Severe disability requiring assistance but can still walk/stand
h. Stage 5.0: Wheelchair-bound or bedridden without assistance
2. Movement Disorder Society-Unified Parkinson's Disease Rating Scale (MDS-UPDRS)<citation>id_3</citation>
a. Four-domain assessment approach
b. Motor and non-motor impact evaluation
c. Replacement for older staging systems
3. Disease Progression Timeline and Transition Indicators <citation>id_16, id_17, id_19</citation>
a. Median transition times between stages (20-26 months)
b. HY > 3 as critical threshold for advanced disease
c. Shift from dopaminergic to cortical pathology involvement
B. Early Stage Warning Signs (Stages 1-2)
1. Motor Symptoms <citation>id_2, id_5, id_7, id_18, id_22</citation>
a. Subtle unilateral tremor (often starting in finger)

b. Micrographia (small handwriting)

c. Bradykinesia (slowed movements)

d. Mild rigidity and stiffness

e. Reduced arm swing

f. Facial masking (reduced expression)

g. Shuffling gait with irregular stride

2. Non-Motor Early Indicators <citation>id_3, id_5, id_7, id_18</citation>

a. Hyposmia (loss of smell)

b. Constipation

c. Sleep disturbances and REM sleep behavior disorder

d. Depression and anxiety

e. Subtle cognitive changes

f. Autonomic dysfunction (blood pressure fluctuations)

g. Unexplained weight loss

3. Diagnostic Challenges and Recognition <citation>id_4</citation>

a. Non-motor signs can appear decades before motor symptoms

b. Overlap with other disorders complicates diagnosis

c. Need for asymmetric symptoms or medication response for confirmation

C. Progressive Stage Warning Signs (Stages 2-3)

1. Advancing Motor Symptoms <citation>id_5, id_6, id_18, id_22</citation>

a. Bilateral tremor and rigidity

b. Postural deterioration and stooped posture

c. Balance problems and impaired reflexes

d. Speech changes (soft, monotone voice)

e. Swallowing difficulties

f. Freezing episodes

2. Emerging Complications <citation>id_16, id_18, id_22</citation>

a. Medication wearing-off periods

b. Levodopa-induced dyskinesia

c. Increased fall risk

d. Emotional and mood changes

3. Functional Impact Assessment <citation>id_6, id_20, id_22</citation>

a. Loss of independence in daily activities

b. Need for assistance with dressing, eating, bathing

c. Occupational and social limitations

D. Advanced Stage Warning Signs (Stages 4-5)

1. Severe Motor Impairment <citation>id_5, id_6, id_9, id_22</citation>

a. Major mobility loss requiring assistive devices

b. Severe postural deformities

c. Inability to live independently

d. Wheelchair or bed confinement

2. Complex Non-Motor Complications <citation>id_5, id_16, id_19, id_22</citation>

a. Cognitive decline and dementia

b. Hallucinations and delusions

c. Severe autonomic dysfunction

d. Chronic pain and sexual dysfunction

3. End-Stage Considerations <citation>id_6, id_9, id_20</citation>

a. Complete dependency for daily activities

b. Risk of aspiration and respiratory complications

c. Need for constant supervision and care

II. Critical Warning Signs Requiring Immediate Medical Intervention

A. Emergency Situations for Family Members

1. Respiratory and Swallowing Emergencies <citation>id_25, id_26, id_29, id_31, id_32, id_35, id_36</citation>

a. Sudden choking or difficulty swallowing

b. Aspiration risk with coughing while eating

c. Severe voice changes or stridor

d. Respiratory difficulty or shortness of breath
2. Neurological Crisis Signs <citation>id_24, id_26, id_28, id_32, id_33, id_35</citation>
a. Sudden confusion, delirium, or marked cognitive decline
b. New or worsening hallucinations, delusions, or paranoid thoughts
c. Abrupt neurological deficits (weakness, speech changes)
d. Loss of consciousness or seizures
3. Medication-Related Emergencies <citation>id_24, id_28, id_33</citation>
a. Abrupt stopping or alteration of medications
b. Exposure to dopamine-blocking drugs
c. Missed or delayed essential medication doses
d. Signs of serotonin syndrome
4. Motor System Crisis <citation>id_25, id_28, id_31, id_34</citation>
a. Sudden severe rigidity or immobility
b. Frequent falls or freezing episodes
c. Rapid loss of walking ability
d. Severe dyskinesia with complications
B. Cardiovascular and Autonomic Emergencies
1. Circulatory Complications <citation>id_26, id_28, id_31, id_32, id_34, id_36</citation>
a. Severe orthostatic hypotension causing fainting
b. Chest pain or cardiac symptoms
c. Signs of blood clots (leg swelling, pain, redness)
d. Marked dehydration
2. Infection and Systemic Issues <citation>id_24, id_26, id_32, id_35</citation>
a. Sudden fever with respiratory or urinary symptoms
b. Signs of urinary tract infection
c. Post-surgical complications
d. Severe constipation or bowel impaction
C. Psychiatric and Behavioral Emergencies
1. Mental Health Crises <citation>id_28, id_32, id_33, id_36</citation>
a. Severe depression or suicidal thoughts
b. Violent behaviors during REM sleep disorder
c. Severe impulse-control behaviors
d. Acute psychosis with agitation or aggression
2. Device-Related Emergencies <citation>id_28, id_34</citation>
a. Sudden malfunction of implanted devices
b. DBS system problems (infection, battery failure)
c. Unexpected sleep attacks during critical activities
D. Hospitalization and Perioperative Risks <citation>id_32, id_35</citation>
1. Hospital-Related Complications
a. Medication timing disruptions
b. Delirium and confusion in unfamiliar environments
c. Increased fall risk and injury potential
d. Aspiration pneumonia during procedures
2. Post-Surgical Warning Signs
a. Rapid symptom deterioration
b. Signs of infection or wound complications
c. Unexpected neuropsychiatric changes
III. Deep Brain Stimulation (DBS) Surgery: Comprehensive Care Framework
A. Pre-Operative Considerations and Evaluation
1. Candidate Selection Criteria <citation>id_45, id_56</citation>
a. Minimum four years of Parkinson's disease
b. Continued medication benefit with motor complications
c. Absence of dementia or severe cognitive impairment
d. Realistic expectations about outcomes
2. Pre-Surgical Assessment Process <citation>id_39, id_45, id_56</citation>
a. Multidisciplinary team evaluation
b. Neurological and neurosurgical consultation

c. Cognitive testing and brain imaging

d. Medication review and optimization

3. Target Selection Considerations <citation>id_53</citation>

a. Subthalamic nucleus (STN) for greater medication reduction

b. Globus pallidus interna (GPi) for language/cognition preservation

c. Individual patient factors influencing choice

B. Surgical Procedure and Immediate Post-Operative Care

1. Surgical Process <citation>id_45, id_49, id_51, id_56</citation>

a. Electrode implantation in target brain regions

b. Pulse generator placement under collarbone

c. Brief hospital stay (1-2 days)

d. Low mortality rate (<0.5%) and modest complication rates (4-7%)

2. Initial Recovery Phase <citation>id_41, id_47, id_49, id_51, id_52, id_65</citation>

a. Expected post-operative signs (bruising, swelling, tenderness)

b. Temporary "honeymoon" or microlesion effect

c. Activity restrictions (4-6 weeks)

d. Wound care and infection prevention

3. Early Post-Operative Monitoring <citation>id_65, id_88</citation>

a. Incision care and healing assessment

b. Medication resumption protocols

c. Activity limitation guidelines

d. Signs requiring immediate medical attention

C. Device Programming and Optimization

1. Programming Timeline <citation>id_41, id_45, id_46, id_47, id_51, id_56</citation>

a. Initial activation 2-4 weeks post-surgery

b. Optimization period of 4-6 months

c. Multiple programming visits required

d. Ongoing adjustments every 6 months

2. Expected Outcomes and Adjustments <citation>id_46, id_51, id_53, id_56</citation>

a. Significant motor symptom improvement (= 32%)

b. Medication reduction (up to 48%)

c. Functional independence gains

d. Possible side effects requiring management

3. Programming Process and Patient Education <citation>id_80, id_86, id_90</citation>

a. Patient-controlled adjustments with handheld programmer

b. Battery monitoring and replacement scheduling

c. Understanding normal sensations vs. concerning symptoms

d. Communication protocols with programming team

IV. Daily Life Adjustments and Support Strategies for DBS Patients

A. Physical Care and Safety Measures

1. Activity Modifications <citation>id_47, id_51, id_52, id_65, id_67</citation>

a. Gradual resumption of normal activities

b. Continued use of mobility aids to prevent falls

c. Avoidance of high-risk activities (deep water diving, extreme heat)

d. Regular exercise program (minimum 2.5 hours weekly)

2. Device Safety and Maintenance <citation>id_51, id_52, id_69, id_77, id_88</citation>

a. Protection from electromagnetic interference

b. Carrying handheld controller at all times

c. Informing healthcare providers about implanted device

d. Regular battery monitoring and replacement

3. Environmental Safety Considerations <citation>id_88, id_90</citation>

a. Contraindicated therapies and equipment

b. Safe vs. unsafe daily activities

c. Travel and security considerations

d. Workplace and recreational restrictions

B. Medication Management Post-DBS

1. Immediate Post-Operative Medication Protocol <citation>id_84, id_88, id_93</citation>

a. Continuation of pre-surgical regimen initially
b. Gradual, systematic reduction approach
c. Monitoring for withdrawal symptoms
d. Coordination with stimulation programming
2. Long-Term Medication Optimization <citation>id_83, id_84, id_85, id_93</citation>
a. Levodopa dosing adjustments
b. Dopamine agonist tapering strategies
c. Management of persistent symptoms
d. Monitoring for mood and cognitive changes
3. Troubleshooting Medication Issues <citation>id_81, id_86</citation>
a. Home-health nurse medication reviews
b. Emergency protocols for medication disruption
c. Balancing stimulation with pharmaceutical needs
d. Communication with neurology team
C. Home-Based Care and Comfort Measures
1. Daily Routine Optimization <citation>id_63, id_65, id_67, id_72</citation>
a. Structured daily routines and symptom diaries
b. Sleep hygiene and rest protocols
c. Regular physical activity programs
d. Comfort interventions (massage, voice training)
2. Home Health Services <citation>id_72, id_81</citation>
a. Trained home-health nurse visits
b. Remote programming capabilities
c. Vital sign monitoring and medication adjustments
d. Reduced travel burden for families
3. Device Troubleshooting at Home <citation>id_77, id_78, id_86, id_90</citation>
a. Basic device operation and monitoring
b. Recognition of malfunction signs
c. Emergency contact protocols
d. Battery status management
D. Psychosocial Support and Adaptation
1. Identity and Role Adjustment <citation>id_54, id_66, id_68</citation>
a. Managing "biographical disruption" from rapid symptom relief
b. Negotiating new caregiver-patient role boundaries
c. Addressing "burden of normality" phenomenon
d. Reclaiming personal interests and autonomy
2. Family Relationship Dynamics <citation>id_54, id_58, id_66, id_68</citation>
a. Shifting from caregiver-patient to balanced spousal relationships
b. Reducing over-protective behaviors
c. Open communication about changing abilities
d. Professional counseling and cognitive-behavioral therapy
3. Caregiver Support and Education <citation>id_61, id_66, id_68</citation>
a. Pre-operative education and expectation setting
b. Structured psychosocial interventions
c. Self-help groups and peer support networks
d. Caregiver self-care and stress management
E. Comprehensive Support Systems
1. Healthcare Team Coordination <citation>id_39, id_46, id_61, id_72</citation>
a. Multidisciplinary follow-up care
b. Specialized Parkinson's nurses
c. Home-health nursing services
d. Telemedicine and remote programming options
2. Long-Term Quality of Life Optimization <citation>id_51, id_54, id_61</citation>
a. Patient satisfaction monitoring (95% would recommend DBS)
b. Positive psychology interventions
c. Character-strength exercises
d. Long-term relationship and social support maintenance

3. Emergency Preparedness and End-of-Life Considerations <citation>id_78</citation>
a. Off-trial protocols for device assessment
b. Palliative care coordination
c. Hardware monitoring in advanced stages
d. Family decision-making support
F. Practical Implementation Strategies
1. Daily Living Skills <citation>id_60, id_63, id_67</citation>
a. Gradual re-engagement in routine tasks
b. Nutritional support (Mediterranean diet)
c. Intellectual and social activities
d. Family encouragement and participation
2. Long-term Monitoring and Adjustment <citation>id_47, id_52, id_69</citation>
a. Regular follow-up visits (monthly for first six months)
b. Symptom tracking and device optimization
c. Monitoring for complications or side effects
d. Battery replacement planning (every 3-5 years)
3.    Resource Utilization and Support Networks <citation>id_55, id_58, id_76, id_79</citation>
a. Educational materials and patient guides
b. Webinars and professional consultations
c. Support groups and peer networks
d. Specialized healthcare provider access
**</write_outline>**

| Step 8 |
|---|

**<think>**
I have now created a comprehensive, detailed outline that addresses all aspects of the user's query. The outline includes:
1. Detailed Parkinson's disease staging and progression with comprehensive warning signs
2. Specific emergency situations requiring immediate medical intervention
3. Complete DBS surgery information including pre-operative, operative, and post-operative care
4. Extensive daily life adjustments and support strategies for DBS patients
5. Proper citations for all subsections

The outline is well-structured with four levels of hierarchy and covers all essential aspects requested. It provides practical, actionable information for families caring for Parkinson's patients, particularly those who have undergone DBS surgery. The outline is comprehensive, logically organized, and includes insights into the reasoning behind various care strategies.
I believe this outline now fully addresses the user's comprehensive query about Parkinson's disease warning signs, family alert systems, and post-DBS care strategies.
**</think>**
<terminate> ✓

## I.7 CASE STUDY OF WRITING

We present only the first 7 steps of the long writing trajectory to save space. The afterwards reasoning process is similar to that in the first 7 steps.

| Case Trajectory of Writing in DeepResearch Bench |
|---|

**Question:** Parkinson's disease has a profound impact on patients. What are the potential health warning signs associated with different stages of the disease? As family members, which specific signs should alert us to intervene or seek medical advice regarding the patient's condition? Furthermore, for patients who have undergone Deep Brain Stimulation

(DBS) surgery, what daily life adjustments and support strategies can be implemented to improve their comfort and overall well-being?

**Step 1**

<think>
I need to write a comprehensive article about Parkinson's disease warning signs, emergency situations for family members, and post-DBS care strategies. Looking at the outline, I should start by retrieving information about disease staging systems and warning signs. Let me begin with the first section on understanding Parkinson's disease progression and warning signs.
</think>

<tool_call>
{"name": "retrieve", "arguments": {"url_id": ["id_2", "id_6", "id_9", "id_12", "id_13", "id_14", "id_15", "id_17", "id_20", "id_21"], "goal": "Gather comprehensive information about Hoehn and Yahr scale classifications and disease staging systems for Parkinson's disease"}}
</tool_call>

<tool_response>
The page content for the previous section has been masked for saving the space.
</tool_response>

**Step 2**

<think>
I have comprehensive information about the Hoehn and Yahr scale classifications and disease staging systems. This provides a solid foundation for understanding Parkinson's disease progression. I can see the clear progression from Stage 0 (no disease) through Stage 5 (wheelchair/bedridden), with intermediate stages 1.5 and 2.5 in the modified version. The information shows both motor and non-motor symptoms, timing of transitions between stages, and functional impacts. I should now write the takeaway section and begin the first major section on disease staging systems.
</think>

<write>
<takeaway>
Parkinson's disease progresses through distinct stages with recognizable warning signs that families must understand to provide appropriate care and seek timely medical intervention. The disease advances from subtle unilateral symptoms in early stages to severe bilateral impairment requiring constant care in advanced stages. Critical emergency signs include sudden choking, severe falls, medication disruptions, neurological changes, and psychiatric crises that demand immediate medical attention. For patients who have undergone Deep Brain Stimulation (DBS) surgery, comprehensive daily life adjustments encompass structured post-operative care, medication management, device safety protocols, psychosocial support, and coordinated family involvement to optimize comfort and long-term outcomes. Understanding these progressive warning signs and implementing evidence-based support strategies enables families to navigate the complex challenges of Parkinson's disease while maintaining patient dignity and quality of life.
</takeaway>

Parkinson's Disease: Health Warning Signs, Family Alert Systems, and Post-DBS Care Strategies

I. Understanding Parkinson's Disease Progression and Warning Signs

A. Disease Staging Systems

Understanding Parkinson's disease progression requires familiarity with established staging systems that provide frameworks for recognizing symptom evolution and functional decline. These systems serve as critical tools for families to understand what to expect and when to seek medical intervention.

1. Hoehn and Yahr Scale Classifications

The Hoehn and Yahr scale, originally published in 1967, remains the most widely used system for staging Parkinson's disease progression. <cite id="id_14,id_17">This scale distinguishes between five stages of Parkinson's Disease, from unilateral impairment (Stage 1) to bilateral impairment, without postural control difficulties (Stage 2), to postural instability

(Stage 3), to loss of physical independence (Stage 4) and finally, to wheelchair-bound or bed-bound (Stage 5)</cite>. The modified version includes intermediate stages that capture more nuanced progression patterns.
<table>
<caption>Hoehn and Yahr Scale Progression and Warning Signs</caption>
<thead>
<tr>
<th>Stage</th>
<th>Primary Characteristics</th>
<th>Functional Impact</th>
<th>Key Warning Signs for Families</th>
</tr>
</thead>
<tbody>
<tr>
<td>Stage 0</td>
<td><cite id="id_12,id_21">No signs of disease</cite></td>
<td>Normal function</td>
<td>Baseline monitoring period</td>
</tr>
<tr>
<td>Stage 1.0</td>
<td><cite id="id_15,id_20">Unilateral tremor, rigidity, reduced arm swing, and slowness present only on one side</cite></td>
<td><cite id="id_2,id_6">Minimal interference with daily activities</cite></td>
<td>Subtle tremor in one finger, asymmetric arm swing</td>
</tr>
<tr>
<td>Stage 1.5</td>
<td><cite id="id_15,id_21">Unilateral involvement with axial symptoms like reduced facial expression, stooped posture, or reduced arm swing on both sides</cite></td>
<td>Slight postural changes</td>
<td>Facial masking, posture changes visible to family</td>
</tr>
<tr>
<td>Stage 2.0</td>
<td><cite id="id_13,id_17">Bilateral involvement without impairment of balance</cite></td>
<td><cite id="id_2,id_6">Daily activities become more difficult but independent living realistic</cite></td>
<td>Tremor and stiffness on both sides, speech difficulties</td>
</tr>
<tr>
<td>Stage 2.5</td>
<td><cite id="id_15,id_21">Mild bilateral disease with recovery on pull test (taking $\geq 3$ steps to recover balance)</cite></td>
<td>Mild balance impairment</td>
<td>Slight unsteadiness, requires multiple steps to recover balance</td>
</tr> <tr> <td>Stage 3.0</td>
<td><cite id="id_13,id_20">Mild to moderate bilateral disease with impaired postural reflexes; physically independent</cite></td>
<td><cite id="id_2,id_6">Falls become frequent, assistance needed for some activities</cite></td> <td>Loss of balance when pulled backward, increased fall risk</td>
</tr> <tr> <td>Stage 4.0</td>
<td><cite id="id_13,id_20">Severe disability; still able to walk or stand unassisted but markedly incapacitated</cite></td>
<td><cite id="id_2,id_6">Cannot live independently, requires walker or assis-

tance</cite></td> <td>Need for assistive devices, inability to live alone safely</td>
</tr> <tr> <td>Stage 5.0</td>
<td><cite id="id_13,id_21">Wheelchair bound or bedridden unless aided</cite></td>
<td><cite id="id_2,id_6">Around-the-clock nursing care required, up to 50% experience hallucinations and delusions</cite></td>
<td>Inability to stand or walk, cognitive disturbances, complete dependency</td> </tr>
</tbody>
</table>

This staging system reveals several critical insights for families. **The transition from Stage 2 to Stage 3 marks a pivotal point** where balance problems emerge, fundamentally changing the patient's safety profile and care needs. <cite id="id_20">Loss of balance, with the inability to make the rapid, automatic and involuntary movements necessary to protect against falling, is one of the most troubling and dangerous aspects of Parkinsonism and one of the least easily treated</cite>. Understanding this progression helps families anticipate when environmental modifications and increased supervision become essential.

**The distinction between Stage 3 and Stage 4 centers on independence**. <cite id="id_20">The most important factor identifying Stage III (as opposed to stage IV) is that the patient is still fully independent in all activities of daily living (dressing, hygiene, eating, etc.), while Stage IV patients are unable to lead an independent life because of the need for help with some activities of daily living</cite>. This transition signals when families must consider more intensive caregiving arrangements or assisted living options.
**</write>**

| Step 3 |
| --- |

**<think>**
Now I need to continue with information about the MDS-UPDRS system and disease progression timeline. Let me retrieve more specific information about these aspects.
**</think>**
**<tool_call>**
{"name": "search", "arguments": {"url_id": ["id_3", "id_16", "id_17", "id_19"], "goal": "Gather information about MDS-UPDRS assessment system, disease progression timeline, and transition indicators between stages"}}
**</tool_call>**
**<tool_response>**
The page content for the previous section has been masked for saving the space.
**</tool_response>**

| Step 4 |
| --- |

**<think>**
The tool response provides valuable information about the MDS-UPDRS system, disease progression timeline, and the critical transition point at Stage 3. I can see that:
1. MDS-UPDRS has replaced the older Hoehn-Yahr system as the main assessment tool
2. The progression timeline shows median transition times between stages (20-26 months typically)
3. Stage 3 (HY > 3) represents a critical threshold marking advanced disease with increased disability, medication-refractory features, and cognitive decline
4. There's important information about non-motor symptoms appearing years before motor symptoms
5. Specific progression warning signs are detailed
I should continue writing about the MDS-UPDRS system and progression timeline, then move on to early stage warning signs.
**</think>**
**<tool_call>**
{"name": "search", "arguments": ["DBS patient daily living support family caregivers", "Parkinson DBS lifestyle modifications comfort strategies", "DBS programming adjustment patient family guide"], "goal": "Find specific daily life support strategies, comfort measures, and family guidance for caring for Parkinson's patients who have undergone DBS surgery"}}
**</tool_call>**
**<write>**

2. Movement Disorder Society-Unified Parkinson's Disease Rating Scale (MDS-UPDRS)

While the Hoehn and Yahr scale remains valuable for understanding disease progression, <cite id="id_3">the Movement Disorder Society-Unified Parkinson's Disease Rating Scale (MDS-UPDRS) is healthcare providers' main tool to classify this disease. The MDS-UPDRS examines four different areas of how Parkinson's disease affects you</cite>. This comprehensive assessment tool provides a more nuanced evaluation of both motor and non-motor symptoms, addressing the limitations of purely stage-based approaches.

**The significance of this shift toward individualized assessment cannot be overstated**. Rather than simply categorizing patients into stages, the MDS-UPDRS recognizes that Parkinson's disease affects each person differently, requiring personalized treatment approaches. This evolution in assessment methodology reflects our growing understanding that the disease's impact extends far beyond motor symptoms.

3. Disease Progression Timeline and Transition Indicators

Understanding the temporal aspects of Parkinson's progression helps families prepare for changing care needs. <cite id="id_17">Median transition times between stages show Stage 2 progression occurring around 20 months, with subsequent stages progressing every 24-26 months</cite>. However, <cite id="id_16">for some people, it can take as long as 20 years to move through all the stages</cite>, emphasizing the highly variable nature of disease progression.

**The transition to Stage 3 represents a critical threshold** in Parkinson's disease progression. <cite id="id_19">Transitions from scores of $\leq 2.5$ to scores of $\geq 3$ on the modified HY scale mark a worsening in quality of life and an increase in the risk of medication-refractory disease features such as dementia and mortality</cite>. This transition point signals when families must significantly intensify their care planning and support strategies.

The underlying pathophysiology also changes at this critical juncture. <cite id="id_19">Although striatal dopaminergic neuropathology accounts for many of the early motor features of PD, in vivo imaging studies suggest floor effects to striatal dopaminergic denervation that occur both in early PD and at HY stage 3 disease, at which point progressive disease burden is driven by other factors</cite>. This shift from primarily dopaminergic to multi-system pathology explains why treatment becomes more complex and why families observe new types of symptoms emerging.

B. Early Stage Warning Signs (Stages 1-2)

Early recognition of Parkinson's disease symptoms enables timely medical intervention and better long-term outcomes. The challenge for families lies in distinguishing between normal aging changes and early disease manifestations.

1. Motor Symptoms

The classic motor symptoms of Parkinson's disease typically appear gradually and asymmetrically. <cite id="id_3">Motor symptoms — which means movement-related symptoms — of Parkinson's disease include tremor at rest, bradykinesia (slowness of movement), rigidity (muscle stiffness), and postural instability (balance problems)</cite>.

**Tremor characteristics provide important diagnostic clues**. The typical Parkinson's tremor occurs at rest, often beginning subtly in one finger or hand. Families should note that this tremor typically diminishes during purposeful movement, distinguishing it from other types of tremors. The asymmetric onset is particularly significant—symptoms beginning on one side of the body are more suggestive of Parkinson's disease than bilateral symptoms.

**Bradykinesia manifests in multiple ways** that families can observe in daily activities. This slowness of movement affects not only gross motor activities but also fine motor skills, leading to characteristic changes in handwriting (micrographia), reduced facial expression (facial masking), and decreased arm swing during walking. These changes often develop so gradually that they may be attributed to normal aging until they become more pronounced.

**Rigidity and postural changes** represent another category of early motor symptoms. Families may notice increased muscle stiffness, particularly in the neck and shoulders, leading to complaints of aching or discomfort. The development of a stooped posture and reduced arm swing during walking are early indicators that warrant medical evaluation.

2. Non-Motor Early Indicators

Perhaps most significantly for early detection, <cite id="id_3">many of the possible non-motor symptoms can appear years or even decades ahead of motor symptoms</cite>.

This extended prodromal period offers families crucial opportunities for early intervention, though the subtlety of these symptoms often leads to delayed recognition.

**Olfactory dysfunction represents one of the earliest detectable changes**. <cite id="id_3">Loss of sense of smell (anosmia)</cite> occurs in approximately 90% of Parkinson's patients, often years before motor symptoms appear. Families should take note when a loved one reports that food no longer tastes the same or when they fail to detect strong odors that others notice readily.

**Gastrointestinal symptoms, particularly constipation**, frequently precede motor symptoms by years. <cite id="id_3">Constipation</cite> in Parkinson's disease results from the same pathological processes affecting the brain, specifically involving the enteric nervous system. This symptom is often dismissed as age-related or dietary, but persistent constipation without clear cause warrants medical attention.

**Sleep disturbances provide another early warning system**. <cite id="id_3">Rapid eye movement sleep behavior disorder</cite> is particularly significant, as it involves acting out dreams during sleep, potentially including violent movements or vocalizations. <cite id="id_16">REM sleep behavior disorder can also happen in people who don't have Parkinson's. However, if this isn't something you've dealt with before, it's likely related to your disease</cite>. Family members sharing a bed are often the first to notice these changes.

**Neuropsychiatric symptoms deserve particular attention** from families. <cite id="id_3">Depression or anxiety</cite> can appear years before motor symptoms, and <cite id="id_16">anxiety and depression have been linked to Parkinson's. In addition to movement problems, the disease can also have an impact on your mental health</cite>. These mood changes may be subtle initially but can significantly impact quality of life and relationships.
**</write>**

| Step 5 |
|---|

**<think>**

Now I need to continue with information about progressive stage warning signs and advanced stage warning signs. Let me retrieve more specific information about these later stages and their complications.
**</think>**

**<tool_call>**
{"name": "retrieve", "arguments": {"url_id": ["id_5", "id_6", "id_18", "id_22"], "goal": "Gather detailed information about progressive and advanced stage warning signs, including motor and non-motor complications"}}
**</tool_call>**

**<tool_response>**
The page content for the previous section has been masked for saving the space.
**</tool_response>**

| Step 6 |
|---|

**<think>**

The tool response provides excellent detailed information about progressive and advanced stage warning signs. I can see clear patterns in the progression from mild to moderate to advanced stages, with specific motor and non-motor symptoms at each stage. The information shows how symptoms evolve from unilateral to bilateral, how functional independence is progressively lost, and how non-motor symptoms become increasingly prominent and disabling. I should organize this information to show the progression clearly and highlight the key warning signs families need to recognize.
**</think>**

**<write>**

C. Progressive Stage Warning Signs (Stages 2-3)

As Parkinson's disease advances beyond the early stage, families observe more pronounced symptoms that begin to significantly impact daily functioning. This progressive phase represents a critical period where proactive intervention can substantially influence quality of life outcomes.

1. Advancing Motor Symptoms

**The bilateral spread of symptoms marks a fundamental shift** in disease progression.

<cite id="id_18">As the disease progresses, it eventually affects both sides. However, the symptoms may still be more severe on one side than on the other</cite>. This bilateral involvement signals that families must adapt their support strategies to address more comprehensive functional limitations.

<cite id="id_22">Often within 3 to 7 years, you'll see more changes. Early on, you might have a little trouble with something like buttoning a shirt. At this point, you may not be able to do it at all</cite>. **This progression from mild difficulty to complete inability** represents a pattern families will observe across multiple daily activities, requiring gradual assumption of caregiving responsibilities.

**Speech and swallowing changes become particularly noticeable** during this stage. <cite id="id_5">You may speak softly or quickly, slur, or hesitate before talking. Your speech may be flat or monotone, without typical speech patterns</cite>. These changes affect not only communication but also safety, as <cite id="id_22">trouble swallowing</cite> increases the risk of aspiration and nutritional deficiencies.

**Gait disturbances and freezing episodes** represent some of the most challenging aspects of progressive Parkinson's disease. <cite id="id_18">People with Parkinson's disease often develop a parkinsonian gait that includes a tendency to lean forward; take small, quick steps; and reduce swinging their arms. They also may have trouble initiating or continuing movement</cite>. The emergence of <cite id="id_22">freezing when you first start to walk or change direction, as if your feet are glued to the ground</cite> creates significant safety concerns and mobility limitations.

2. Emerging Complications

**Medication effectiveness begins to fluctuate** during the progressive stage, creating new challenges for symptom management. <cite id="id_22">You might also find that the medicine you take starts to wear off between doses</cite>. This "wearing-off" phenomenon requires families to become more attentive to timing of medications and to recognize when symptoms return between doses.

**Balance problems and fall risk increase substantially**. <cite id="id_6">At this stage, falls become more frequent due to balance and reflex problems</cite>, and <cite id="id_22">falls to be more likely</cite>. This increased fall risk necessitates environmental modifications and potentially the introduction of assistive devices.

3. Functional Impact Assessment

**The distinction between Stages 2 and 3 centers on independence preservation**. <cite id="id_6">Due to the various issues this stage presents, daily tasks are much more difficult to do, but people can still live independently. A combination of medication and therapy can help to manage the symptoms mentioned above</cite>. Understanding this threshold helps families recognize when to increase support while still encouraging independence.

D. Advanced Stage Warning Signs (Stages 4-5)

The advanced stages of Parkinson's disease present complex challenges that require comprehensive care planning and family coordination. These stages are characterized by severe functional limitations and the emergence of complications that significantly impact quality of life.

1. Severe Motor Impairment

**Stage 4 marks the loss of independent living capability**. <cite id="id_6">The critical factor in separating people with stage 3 Parkinson's and stage 4 is independence. Motor skills and deep brain stimulation are heavily impacted at stage 4 and as such, movement symptoms affect a person's ability to retain their independence</cite>. This transition requires families to make difficult decisions about living arrangements and care provision.

<cite id="id_22">At this point, you likely: Are limited to bed or a wheelchair, Can't live on your own, Have severe posture issues in your neck, back, and hips, Need help with daily tasks</cite>. **The comprehensive nature of these limitations** means that families must coordinate multiple aspects of care, from basic hygiene to mobility assistance.

**Stage 5 represents the most severe manifestation** of motor symptoms. <cite id="id_6">Severe stiffness can make it difficult, if not impossible, for a person to stand or walk. This is due to stiffness causing the legs to essentially freeze when the patient attempts to stand</cite>. This "freezing" phenomenon at the advanced stage differs qualitatively from earlier freezing episodes, representing a more permanent limitation rather than intermittent

difficulty.

2. Complex Non-Motor Complications

**Cognitive decline becomes a prominent feature** in advanced stages. <cite id="id_18">Over time, as the disease progresses, some people may develop dementia and be diagnosed with Parkinson's dementia, a type of Lewy body dementia. People with Parkinson's dementia may have severe memory and thinking problems that affect daily living</cite>. This cognitive impairment adds another layer of complexity to caregiving responsibilities.

**Psychiatric symptoms emerge as significant challenges**. <cite id="id_6">Furthermore, up to 50% of Parkinson's patients in stages 4 and 5 experience confusion, hallucinations, and delusions. For clarity, hallucinations are seeing things that aren't really there, and delusions are when a person believes something despite evidence to the contrary</cite>. These symptoms can be particularly distressing for families and may require specialized psychiatric intervention.

**Swallowing and nutritional complications** become life-threatening concerns. <cite id="id_5">Late-stage Parkinson's disease affects the muscles in the mouth. This causes trouble swallowing and chewing, which can lead to not getting enough nutrients in your diet. If food or saliva collects in the mouth, it can cause choking or drooling</cite>. These complications require careful monitoring and may necessitate alternative feeding methods.

<table>
<caption>Progressive Warning Signs by Disease Stage</caption>

| Stage | Motor Warning Signs | Non-Motor Warning Signs | Functional Impact | Family Action Required |
|---|---|---|---|---|
| Early (1-2) | Unilateral tremor, mild rigidity, reduced arm swing, micrographia | Sleep disturbances, constipation, loss of smell, mood changes | Minimal daily life disruption | Medical evaluation, symptom monitoring |
| Progressive (2-3) | Bilateral symptoms, gait freezing, speech changes, balance problems | Medication wearing-off, cognitive changes, swallowing difficulty | Increased difficulty with daily tasks, still independent | Environmental modifications, fall prevention, medication timing |
| Advanced (4-5) | Severe rigidity, wheelchair/bed bound, postural deformities | Dementia, hallucinations, severe dysphagia, autonomic failure | Complete dependency for daily activities | 24-hour care, safety monitoring, end-of-life planning |

</table>

**The temporal progression of non-motor symptoms** follows a predictable pattern that families should understand. <cite id="id_22">What may show up early. You may have these issues years before any classic motor symptoms like tremors: Constipation, Depression, Loss of smell, Low blood pressure when you stand up, Pain, Sleep issues</cite>. Conversely, <cite id="id_22">What may show up later. Dementia and psychosis are two serious mental health issues that usually take a while to show up</cite>.

This progression pattern reveals that **non-motor symptoms often become more disabling than motor symptoms** in advanced stages. <cite id="id_22">Almost everyone with Parkinson's gets at least one of these. When severe, they're more likely than motor issues to lead to a disability or make you move into a nursing home</cite>. This insight helps families understand why comprehensive care planning must address both motor and non-motor aspects of the disease.

3. End-Stage Considerations

**The advanced stage requires families to confront difficult realities** about prognosis and care goals. <cite id="id_22">After about 10 years, most people will have at least one major issue, like dementia or a physical disability</cite>. Understanding this timeline helps families engage in advance care planning while the patient can still participate in decision-making. **The variability in progression** means that some individuals may never reach the most severe stages, while others progress more rapidly. <cite id="id_22">Some people never reach this stage</cite> of advanced disability, emphasizing the importance of individualized care planning rather than assuming inevitable decline. **</write>**

| Step 7 |
|---|

**<think>**
Now I need to move on to the second major section about critical warning signs requiring immediate medical intervention. This is crucial information for families to know when to seek emergency care. Let me retrieve information about emergency situations and warning signs. **</think>**
**<tool_call>**
{"name": "retrieve", "arguments": {"url_id": ["id_24", "id_25", "id_26", "id_28", "id_29", "id_31", "id_32", "id_33", "id_34", "id_35", "id_36"], "goal": "Gather comprehensive information about emergency warning signs and situations that require immediate medical intervention for Parkinson's patients"}}
**</tool_call>**
**<tool_response>**
The page content for the previous section has been masked for saving the space.
**</tool_response>**

| Step 8 |
|---|

...
<terminate> ✓