# OpenReview forum: "WebWeaver: Structuring Web-Scale Evidence with Dynamic Outlines for Open-Ended Deep Research"
_ICLR.cc/2026/Conference — ICLR 2026 Poster_

### Official Review · Reviewer_W3Xh · 2025-10-31

**Soundness:** 3
**Presentation:** 3
**Contribution:** 3
**Rating:** 6
**Confidence:** 4

**Summary:**

The paper proposes WebWeaver, a dual-agent “planner + writer” framework for open-ended deep research. The planner dynamically produces a structured outline with linked evidence; the writer then retrieves the required evidence section by section and composes the content hierarchically based on the outline. The authors conduct experiments and ablations across multiple benchmarks, and the results demonstrate the effectiveness of the approach.

**Strengths:**

1. The paper is easy to follow, the figures are clear, and the experimental details are thorough. The final results look solid.

2. It folds “dynamic outline optimization + evidence binding” into the planning loop and uses section-wise retrieval + writing, which mirrors how humans actually do deep research.

3. It releases a 3k SFT dataset and shows gains on smaller models, which supports the method’s effectiveness.

**Weaknesses:**

1. The planning stage parses 100+ pages and burns tens of thousands of tokens, but there’s no apples-to-apples report of wall-clock latency or API cost vs. baselines.
2. It is recommended to include parameter-matched comparisons across different frameworks built on the same model backbone, or reproduction experiments against other strong baselines on the identical model.

**Questions:**

See weakness

---

> ### Author Response · Authors · 2025-11-20
> **Response to Reviewer W3Xh (1/2)**
>
> > #### W1. The planning stage parses 100+ pages and burns tens of thousands of tokens, but there’s no apples-to-apples report of wall-clock latency or API cost vs. baselines.
>
>
> We thank the reviewer for the comment regarding costs.  We provide a detailed cost analysis in Table R16. We must first note that wall-clock times are highly sensitive to external factors like API latency, network conditions, and concurrent GPU capabilities. As our experiments were conducted under resource-constrained academic conditions (8 * H20 GPU), the reported times should be viewed as a practical reference rather than an absolute benchmark. With that context, our primary model using Claude-sonnet-4 is the most resource-intensive (e.g., ~29 min and $2.82 per task on DeepResearch Bench). This higher cost is a direct consequence of its comprehensive evidence gathering, which enables the high-quality reports for writing.
>
> To provide a more efficient alternative, our fine-tuned model, Qwen3-30b-a3b, drastically reduces latency for planning and writing. Notably, the total end-to-end time ( ~ 18 min on DeepResearch Bench) is highly competitive with the Gemini baseline's estimated generation time (~16 min). This demonstrates that our framework is both practical and efficient even under academic research constraints. We have included detailed discussions about costs in the revised manuscript.
>
>
>
> |    |   | Planner  |  |  | Writer  |  |   | Search API Call |     |
> | -- | -- | --| -- | -- | --- | -- | --- | -- | --- |
> |  | Input token | Output token | Time  | Input token | Output token | Time | Money of LLM API | Google Search | ScraperAPI |
> | DeepResearch Bench  |  |   | | | | | |||
> | Gemini-2.5-pro-deepresearch | -   | -  | -  | - | - | ~16 min  | -   | -  | -   |
> | WebWeaver (Claude-sonnet-4-20250514) | 196,808   | 10,298  | 13.4 min | 522,765  | 33,653  | 15.6 min | $2.82   | 20.24  | 112.25  |
> | WebWeaver (Qwen3-30b-a3b)   | 163,510 | 8,447   | 11.8 min | 564,676 | 36,619  | 6.00 min | -  | 16.65  | 110.23  |
> | DeepConsult  |  |  |  |  |  | |  |  | |
> | Gemini-2.5-pro-deepresearch  | -  | - | - | -  | -  | ~14 min  | -  | - | -  |
> | WebWeaver (Claude-sonnet-4-20250514) | 188,084     | 9,045  | 12.5 min | 471,929  | 29,525  | 14.8 min | $2.56 | 17.91 | 109.86  |
> | WebWeaver (Qwen3-30b-a3b)| 151,381 | 7,933 | 10.1 min | 504,406 | 28,825 | 5.40 min | -  | 12.8  | 106.65 |
> | DeepResearch Gym |  | |  |  |  |  |  |  |  |
> | Gemini-2.5-pro-deepresearch  | - | - | - | -   | - | ~11 min  | -   | - | - |
> | WebWeaver (Claude-sonnet-4-20250514) | 176,907 | 8,511  | 12.2 min | 504,341  | 33,698 | 14.4 min | $2.67  | 21.93 | 102.55  |
> | WebWeaver (Qwen3-30b-a3b)  | 117,091 | 6,767 | 9.30 min | 587,602  | 34,786   | 5.10 min | -  | 13.5  | 102 |
>
> Table R16. Cost analysis of different agent systems. The money costs ($) are computed based on OpenRouter pricing. The wall-clock time for `Gemini-2.5-pro-deepresearch` are manually measured from the Gemini web interface and represent the time for the final report generation. The token and money costs are unknown for `Gemini-2.5-pro-deepresearch`.

---

> ### Author Response · Authors · 2025-11-20
> **Response to Reviewer W3Xh (2/2)**
>
> > #### W2. It is recommended to include parameter-matched comparisons across different frameworks built on the same model backbone, or reproduction experiments against other strong baselines on the identical model.
>
>
>
> Thank you for this suggestion. To precisely isolate the gains attributable to our framework, we have conducted a new set of controlled, end-to-end comparisons, with the results summarized in Table R17. In the first comparison, both WebWeaver and the ReAct baseline were equipped with the identical backbone LLM (Qwen3-235b-a22b), the same set of tools, and operated in the same environment. The only architectural difference was that WebWeaver employed its core **outline optimization** and **hierarchical retrieval and writing** modules, whereas ReAct followed a standard linear thought-action loop to collect information and output the final report. Similarly, we compared our framework against the Claude-research baseline, a proprietary powerful agent system, using the same Claude-sonnet-4 model to further validate our approach.
>
> The results clearly demonstrate the substantial benefits of the WebWeaver framework. As shown in Table R17, when using the exact same backbone LLM, WebWeaver consistently and significantly outperforms the baselines across all benchmarks. For instance, with Qwen3-235b-a22b, our framework boosts the DeepResearch Bench score from 46.16 to 50.80 and the DeepConsult score from 5.68 to 6.90. A similar significant uplift is observed with the Claude-sonnet-4 model. This provides strong and direct evidence that our performance improvements are not merely a function of the underlying LLM's power, but are directly attributable to the structural advantages conferred by our outline optimization and hierarchical retrieval and writing methods.
>
>
>
> |   | DeepResearch Bench | DeepConsut | DeepResearchGym |
> | --- | -- | -- | -- |
> |     | Overall score  | Avg. score | Avg. score  |
> | ReAct (Qwen3-235b-a22b-instruct-2507)  | 46.16 | 5.68  | 86.72           |
> | WebWeaver (Qwen3-235b-a22b-instruct-2507)  | **50.80**   | **6.90**   | **95.57**  |
> | Claude-reserach (Claude-sonnet-4-20250514) | 45.0 | 4.60  | 80.25  |
> | WebWeaver (Claude-sonnet-4-20250514)  | **50.48** | **6.94**   | **96.74**  |
>
> Table R17. Controlled comparisons with varying LLM backbones.

---

> > ### Comment · Reviewer_W3Xh · 2025-11-21
> >
> > Thx to your response, I have no further questions.

---

> > > ### Author Response · Authors · 2025-11-21
> > > **Thank you for your positive feedback and support**
> > >
> > > Thank you for your positive feedback and support for our work. We are very pleased to hear that our revisions have successfully addressed your questions.

---

### Official Review · Reviewer_yb9V · 2025-10-31

**Soundness:** 3
**Presentation:** 3
**Contribution:** 3
**Rating:** 6
**Confidence:** 2

**Summary:**

The paper presents WebWeaver, a dual-agent system for deep research questions. They distinguish their approach from previous methods like “search-then-generate” or “outline-then-write”  by  introducing a planner–writer loop in which the planner iteratively refines a dynamic outline and the writer performs hierarchical writing. They run experiments on DeepResearch Bench, DeepConsult, and DeepResearchGym and showed they outperformed baselines.

**Strengths:**

- The paper addresses an important and and timely problem.

- The dual-agent design  is very tangible and human-inspired very well motivated.

- Comprehensive experiments across three strong benchmarks

- comprehensive ablation studies

**Weaknesses:**

- Missing cost analysis compared to baselines
The paper does not clearly specify when the planner decides to terminate the outline optimization process—what the stopping criteria or thresholds are.

- Several important open-source baselines are missing, such as DeepResearcher, OpenScholar, and open sourced LLM + tool-use systems. Including these would strengthen claims of generality and fairness.

- The WebWeaver-3k dataset is also said to be built from “diverse queries crawled from the web,” yet the crawling criteria, domains, or filters are unspecified.

**Questions:**

- Did you do any analysis on what actiona cause the most problem of answering wrong? for example things like early termination?
-  how were the web queries for WebWeaver-3k crawled—were they topic-balanced or randomly collected?
 - How would webweaver as a dual agent do in terms of cost compared to single agent architechture

---

> ### Author Response · Authors · 2025-11-20
> **Response to Reviewer yb9V (1/3)**
>
> > #### W1. Missing cost analysis compared to baselines
>
>
>
> We thank the reviewer for the question regarding costs. We provide a detailed cost analysis in Table R12. We must first note that wall-clock times are highly sensitive to external factors like API latency, network conditions, and concurrent GPU capabilities. As our experiments were conducted under resource-constrained academic conditions (8 * H20 GPU), the reported times should be viewed as a practical reference rather than an absolute benchmark. With that context, WebWeaver with Claude-sonnet-4 spends more wall-clock time (e.g., ~29 min and $2.82 per task on DeepResearch Bench) compared with the strong baseline Gemini-2.5-pro-deepresearch. This higher cost is a direct consequence of its comprehensive evidence gathering, which enables the high-quality reports for writing.
>
> To provide a more efficient alternative, our fine-tuned model, Qwen3-30b-a3b, drastically reduces latency for planning and writing. Notably, the total end-to-end time ( ~ 18 min on DeepResearch Bench) is highly competitive with the Gemini baseline's estimated generation time (~16 min). This demonstrates that our framework is both practical and efficient even under academic research constraints. We have included detailed discussions about costs in the revised manuscript.
>
>
>
> |    |   | Planner      |  |  | Writer  |  |   | Search API Call |     |
> | ---- | -- | --| -- | -- | --- | -- | --- | -- | --- |
> |      | Input token | Output token | Time  | Input token | Output token | Time  | Money of LLM API | Google Search   | ScraperAPI |
> | DeepResearch Bench   |  |   |   |   |  |  |  |   |  |
> | Gemini-2.5-pro-deepresearch    | -   | -  | -  | -   | -   | ~16 min  | -   | -  | -   |
> | WebWeaver (Claude-sonnet-4-20250514) | 196,808   | 10,298  | 13.4 min | 522,765     | 33,653  | 15.6 min | $2.82   | 20.24  | 112.25  |
> | WebWeaver (Qwen3-30b-a3b)   | 163,510     | 8,447        | 11.8 min | 564,676     | 36,619  | 6.00 min | -                | 16.65           | 110.23     |
> | DeepConsult     |  |    |  |  |  | |  |  |  |
> | Gemini-2.5-pro-deepresearch      | -           | -            | -        | -  | -  | ~14 min  | -  | - | -  |
> | WebWeaver (Claude-sonnet-4-20250514) | 188,084     | 9,045  | 12.5 min | 471,929  | 29,525  | 14.8 min | $2.56 | 17.91 | 109.86  |
> | WebWeaver (Qwen3-30b-a3b)   | 151,381     | 7,933        | 10.1 min | 504,406     | 28,825 | 5.40 min | -  | 12.8  | 106.65     |
> | DeepResearch Gym   |  |      |  |  |  |  |  |  |  |
> | Gemini-2.5-pro-deepresearch  | -  | - | -   | -           | -            | ~11 min  | -   | -   | -          |
> | WebWeaver (Claude-sonnet-4-20250514) | 176,907     | 8,511        | 12.2 min | 504,341     | 33,698   | 14.4 min | $2.67  | 21.93           | 102.55     |
> | WebWeaver (Qwen3-30b-a3b)  | 117,091     | 6,767        | 9.30 min | 587,602     | 34,786   | 5.10 min | -  | 13.5  | 102 |
>
> Table R12. Cost analysis of different agent systems. The money costs ($) are computed based on OpenRouter pricing. The wall-clock time for `Gemini-2.5-pro-deepresearch` are manually measured from the Gemini web interface and represent the time for the final report generation. The token and money costs are unknown for `Gemini-2.5-pro-deepresearch`.
>
>
>
>
>
> > #### W2. The paper does not clearly specify when the planner decides to terminate the outline optimization process—what the stopping criteria or thresholds are.
>
>
>
> Thanks for your comment. The outline optimization process is not governed by a fixed, predefined threshold. Instead, its termination is explicitly made by the planner itself. Specifically, as we introduce in Section 3.2, we include a dedicated **`[Terminate]` action** for the planner. The planner is prompted to invoke this action when it assesses that the current outline is sufficiently comprehensive to address the user's query and that the gathered evidence is adequate for the writer. This dynamic, agent-driven approach is a deliberate design choice, as it is far more flexible than a static stopping criterion (e.g., a fixed number of rounds). It allows the agent to adapt its planning depth to the complexity of the given query—performing more refinement for complex topics and terminating quickly for simpler ones.

---

> ### Author Response · Authors · 2025-11-20
> **Response to Reviewer yb9V (2/3)**
>
> > #### W3. Several important open-source baselines are missing, such as DeepResearcher, OpenScholar, and open sourced LLM + tool-use systems. Including these would strengthen claims of generality and fairness.
>
>
>
> Thank you for suggesting these important open-source baselines. To provide a comprehensive comparison, we have included results from OpenScholar and WebShaper and conducted new experiments. We noted that the open-sourced DeepResearcher model is based on the ReAct paradigm but trains a much smaller model (Qwen2.5-7B). To ensure a more meaningful and fair comparison, we implemented a powerful ReAct baseline using the same strong backbone LLM (Qwen3-235b-a22b) and tools as WebWeaver. This approach provides a direct, apples-to-apples evaluation, isolating the architectural benefits of WebWeaver's **outline optimization** and **hierarchical retrieval and writing** modules.
>
> The results clearly demonstrate WebWeaver's significant advantages. As shown in Table R13, our framework (50.80) substantially outperforms other open-source systems like OpenScholar (25.56) and WebShaper (34.93) on the RACE-Overall score. More critically, in the controlled head-to-head comparison against the powerful ReAct (Qwen3-235b-a22b) baseline, WebWeaver shows consistent and significant gains across all dimensions, boosting the overall score from 46.16 to 50.80. This directly validates that our architectural innovations, particularly our outline optimization and hierarchical writing, are the key drivers of our superior performance, not just the underlying LLM.
>
>
>
> |    | RACE-Overall | Comp.  | Insight   | Inst.     | Read.     |
> | ---- | -- | -- | -- | -- | -- |
> | OpenScholar  | 25.56  | 21.38     | 15.49     | 38.58     | 29.77     |
> | WebShaper (32B)   | 34.93  | 31.58  | 26.17  | 44.81 | 40.38     |
> | ReAct (Qwen3-235b-a22b-instruct-2507) | 46.16| 45.04 | 43.2      | 49.16     | 46.17 |
> | WebWeaver (Qwen3-235b-a22b-instruct-2507) | **50.80**    | **51.45** | **51.39** | **50.26** | **48.98** |
>
> Table R13. Benchmark evaluation on different agent systems.
>
>
>
> > #### W4. The WebWeaver-3k dataset is also said to be built from “diverse queries crawled from the web,” yet the crawling criteria, domains, or filters are unspecified.
>
>
>
> We crawled web pages from a curated list of high-quality English and Chinese forums and communities, specifically targeting knowledge-intensive sources like Reddit.
>
> Our query generation process involved two main stages: We first used pattern matching to extract a large pool of potential queries. This included text from specific HTML tags (e.g., `<h1>`, `<h2>`, post titles) and sentences ending with a question mark, resulting in an initial set of over candidates. We then utilized LLM to filter and refine these candidates into high-quality, open-ended questions. Each candidate was evaluated using a carefully designed prompt that instructed the model to assess it against several criteria:
>
> - Open-endedness: Does the question require a detailed, synthesized report?
> - Clarity: Is the question well-posed and unambiguous?
> - Feasibility: Can evidence be found on the public web?
> - Refinement and Anonymization: If a question has potential, refine it to meet the above standards and remove any personally identifiable information.
>
> Candidates that passed this evaluation (after refinement, if necessary) were retained. A representative example is "What do you think about DeepSeek?" Due to the high costs of filtering and the subsequent trajectory generation, we construct only nearly 3k samples for WebWeaver-3k. We acknowledge that this automated process, while rigorous, cannot fully replicate the nuance and domain-specific depth of questions crafted by human experts. An expert question example of DeepResearch Bench is "Research Topic: Crafting Techniques for Non-Alcoholic Cocktails. Objective: Investigate current non-alcoholic cocktails to discover innovative production methods and formulations." Despite this, our fine-tuned qwen3-30b-a3b model still significantly outperforms the backbone model in benchmarks (see Figure. 12 of the revised manuscript), demonstrating its capable of deconstructing the formidable OEDR task into learnable demonstrations of thinking, searching, and writing.
>
> Furthermore, our dataset is randomly sampling  and topically diverse. The data distribution on ten domains is shown in Table R14, which contains ten domains, including Science, Entertainment, and Politics.
>
>
>
> |       | Technology & Science | Academics & Education | Entertainment | Economics & Business | Sports | Work & Life | Politics | Culture | Society | History |
> | ----- | ---- | --- | -- | ---- | ------ | -- | -- | -- | -- | --- |
> | Ratio | 23.9                 | 16.6                  | 16.4          | 15.4                 | 7.8    | 5.9         | 4.7      | 3.4     | 3.2     | 2.3     |
>
> Table R14. Domain distribution of WeaWeaver-3k.

---

> ### Author Response · Authors · 2025-11-20
> **Response to Reviewer yb9V (3/3)**
>
> > #### Q1. Did you do any analysis on what actiona cause the most problem of answering wrong? for example things like early termination?
>
> Thank you for the question. However, our framework demonstrates consistently strong performance across all benchmarks, suggesting the premise of frequent 'wrong answers' does not apply. **To be specific, on the DeepResearch Bench, where scoring is based on a direct comparison against a reference answer written by a human expert, a score of 50 represents parity with this expert reference.** Our top model achieves 50.80, indicating that our reports are consistently judged as good as, or even superior to, the human standard. This demonstrates the agent framework superiority of WebWeaver.
>
>
>
> > #### Q2. how were the web queries for WebWeaver-3k crawled—were they topic-balanced or randomly collected?
>
> Please refer to the response to W4.
>
>
>
> > #### Q3. How would webweaver as a dual agent do in terms of cost compared to single agent architechture
>
>
>
> Thanks for your question about costs. We have present the cost comparison is presented in Table R15. We acknowledge that our dual-agent architecture, by design, incurs a higher computational cost in terms of token usage and time compared to a standard single-agent approach. This increased cost is a direct and deliberate consequence of outline optimization and hierarchical retrieval and writing modules, which is the core driver of our system's superior performance.
>
> This design allows the planner to focus exclusively on understanding the query's nuances, conducting a more exhaustive exploration, and gathering a comprehensive set of sources. Subsequently, the writer agent can execute a more deliberate, multi-step process of hierarchical retrieval and writing to construct a more coherent and faithful report. The results in Table R15 show that, while a single-agent system like ReAct is faster, our approach's investment in a more thorough process yields a significant leap in quality (e.g., from 46.16 to over 50 on the Overall Score on DeepResearch Bench).
>
> Ultimately, in the domain of report generation, the primary objective is to deliver high-quality, trustworthy, and verifiable reports. Our Webweaver dual-agent design is an intentional investment to meet this high standard, prioritizing superior output. While we recognize that cost efficiency is an ongoing challenge, we believe that investigating methods to integrate these successful architectural principles into a more cost-optimized, potentially single-agent system offers an exciting and critical avenue for subsequent research.
>
>
>
> |                                       | Total input token | Total output token | time      | Overall Score |
> | ------------------------------------- | ----------------- | ------------------ | --------- | ------------- |
> | ReAct (Qwen3-235b-a22b-instruct-2507) | 58,960            | 6,374              | 10.85 min | 46.16         |
> | WebWeaver (Claude-sonnet-4-20250514)  | 719,573           | 43,951             | 29 min    | 50.48         |
>
> Table R15. Cost comparison between dual agents and single agent systems on DeepResearch Bench.

---

### Official Review · Reviewer_4PPn · 2025-11-01

**Soundness:** 3
**Presentation:** 3
**Contribution:** 3
**Rating:** 6
**Confidence:** 4

**Summary:**

The paper introduces **WebWeaver**, a dual‑agent system (Planner + Writer) for “open‑ended deep research” (OEDR). The Planner iteratively interleaves web search and outline optimisation; the Writer then synthesises a long‑form report section‑by‑section using a citation‑grounded memory bank. The authors claim that this human‑inspired workflow eliminates the “loss‑in‑the‑middle” problem, reduces hallucinations, and yields state‑of‑the‑art results on three recently released OEDR benchmarks (DeepResearch Bench, DeepConsult, DeepResearchGym). They also present a 3 k‑example SFT dataset (WebWeaver‑3k) that purportedly enables a 30 B‑scale model to match the performance of much larger proprietary agents.

**Strengths:**

- Clear, human-inspired, end-to-end system design with an explicit evidence chain: planning → evidence extraction into a memory bank → outline with citation IDs → retrieval-grounded writing. The qualitative case study is coherent and illustrates the intended workflow.
- Across three reasonably comprehensive benchmarks, results indicate competitive to SOTA performance. On DeepResearch Bench, the reported citation accuracy (93.37%) and effective citations are particularly strong. On DeepConsult and DeepResearchGym, the win rate and average scores are strong relative to listed systems.
- Ablations address two core hypotheses: (1) iterative outline optimization improves depth/breadth/support and end-to-end scores, and (2) hierarchical, citation-driven writing outperforms brute-force context stuffing. These analyses are aligned with the proposed mechanism of action.
- Useful operational statistics (number of searches, outline size, number of saved pages, evidence tokens) convincingly argue that memory-centric, targeted retrieval is not just helpful but necessary under the reported scale (>100 pages, >60k evidence tokens).
- The SFT dataset and fine-tuning experiments show that the framework can act as a data engine for transferring agentic skills to smaller models, with large gains in citation accuracy. This is a practical contribution for accessibility.

**Weaknesses:**

- Cost and budget parity are not controlled or reported. WebWeaver averages ~100+ pages parsed, ~67k evidence tokens, and ~26k-token outputs. Many baseline systems (especially proprietary) may have stricter rate/token limits. Without a controlled per-task budget (time, API calls, tokens), performance advantages may partly reflect higher spend rather than method.
- Baseline comparability is unclear. WebWeaver is evaluated with multiple powerful base LLMs (Qwen-235B, Claude Sonnet 4), and even when the “agent model” is Claude, a separate model (GPT-oss-120B) is used for URL selection and evidence extraction. This multi-model, auxiliary-LLM design could materially boost performance yet is not mirrored for baselines. It is therefore difficult to isolate method gains from model capability/tooling differences.
- Statistical analysis is thin. No confidence intervals, no statistical tests, and no repeated runs are reported. Improvements on RACE overall scores are modest versus strong baselines (e.g., ~1 point vs Gemini 2.5), and the absence of variance estimates undermines claims of reliable superiority. DeepConsult reports win/tie/lose counts but no significance testing despite pairwise comparisons being amenable to binomial or bootstrap CIs.
- Reliance on LLM-as-judge across all benchmarks creates evaluation circularity risks; although the paper follows “official” judges per benchmark, additional human evaluation on a representative subset would strengthen claims, especially for “insight,” “readability,” and “support.”
- Effective citations metric may be confounded by output length. WebWeaver outputs very long reports; if “Eff. c.” counts raw validated citations, longer outputs can inflate the score. There is no normalization for output length or per-claim basis reported. The very large Eff. c. numbers (200+) relative to baselines raise the possibility that the metric rewards verbosity rather than calibrated evidence use.
- The ablation “hierarchical vs brute-force writing” lacks sufficient detail to be fully persuasive: context limits, chunking strategy for the brute-force baseline, and exact parity of planner outputs across conditions are not specified. Without these, it’s unclear if the baseline is disadvantaged by context overflow or suboptimal chunking rather than an inherent weakness of the paradigm.
- The outline optimization ablation uses samples with three rounds of optimization and then evaluates earlier rounds; however, the prompt template prescribes “at least three” updates but the reported average outline optimization steps are ~2.2. This mismatch between prescribed behavior and observed behavior should be clarified.
- No code, no executable release, and no links to run-time infrastructure are provided. Prompts are included, but the system depends on multiple LLMs, tools, and scraping steps. Critical parameters (stop criteria, deduplication policies, cache behavior, search engine settings, rate limits, tie-breaking rules, etc.) are not fully specified. This reduces reproducibility.
- The WebWeaver-3k dataset creation lacks overlap analysis against evaluation prompts and lacks licensing audits of scraped content. Without an overlap/contamination check, the risk of implicit test exposure cannot be discounted.
- The ethics section is a little generic at the moment and should also address website terms-of-service compliance, robots.txt adherence, or how PII in web pages is handled during evidence extraction and dataset curation. For an agent that crawls and stores web-scale evidence, these omissions are material.

**Questions:**

1. Code/data release: Will you release the full WebWeaver code (planner/writer, tools, memory) and the WebWeaver-3k dataset? If not, what components will be made public and when?
2. Fairness controls: Can you provide a controlled comparison where WebWeaver and a strong baseline agent both use the same backbone LLM and judge, to isolate framework gains? Any cross-judge robustness (e.g., alternate judges or human eval subsets)?
3. Cost/latency: What are average wall-clock time, token counts, and dollar costs per task on each benchmark, for both planning and writing? How does it compare to baselines?
4. Evidence extraction specifics: How are quote spans selected, how is evidence deduplicated/normalized, and what is the granularity for memory entries? How do you handle noisy, contradictory, or paywalled sources?
5. Failure and correction: How do the planner/writer detect and correct citation errors (e.g., broken links, drifted IDs, semantically mismatched evidence)? Any automatic validation?
6. WebWeaver-3k licensing and distribution: What is the licensing status of the dataset, given it is derived from web materials and proprietary teacher outputs? Can it be redistributed?
7. Safety and misuse: What safeguards are in place to prevent the agent from citing low-quality or harmful sources? Do you filter by source credibility or domain allowlists/blacklists?
8. Outline-to-retrieval mapping: How robust is the citation-ID mapping across multiple optimization rounds (IDs added/removed)? How do you maintain referential integrity?

---

> ### Author Response · Authors · 2025-11-20
> **Response to Reviewer 4PPn (1/7)**
>
> > #### W1. Cost and budget parity are not controlled or reported. WebWeaver averages ~100+ pages parsed, ~67k evidence tokens, and ~26k-token outputs. Many baseline systems (especially proprietary) may have stricter rate/token limits. Without a controlled per-task budget (time, API calls, tokens), performance advantages may partly reflect higher spend rather than method.
>
>
>
> We thank the reviewer for this important question regarding costs. We have no evidence to suggest proprietary baselines intentionally limit their token usage to save costs; rather, they normally restrict the experience times and provide the best-effort response for each query.
>
> We provide a detailed cost analysis in Table R4. We must first note that wall-clock times are highly sensitive to external factors like API latency, network conditions, and concurrent GPU capabilities. As our experiments were conducted under resource-constrained academic conditions (8 * H20 GPU), the reported wall-clock time should be viewed as a practical reference rather than an absolute metric. With that context, WebWeaver with Claude-sonnet-4 spends more wall-clock time (e.g., ~29 min and $2.82 per task on DeepResearch Bench) compared with the strong baseline Gemini-2.5-pro-deepresearch. This higher cost is a direct consequence of its comprehensive evidence gathering, which enables the high-quality reports for writing.
>
> To provide a more efficient alternative, our fine-tuned model, Qwen3-30b-a3b, drastically reduces latency for planning and writing. Notably, the total end-to-end time ( ~ 18 min on DeepResearch Bench) is highly competitive with the Gemini baseline's estimated generation time (~16 min). This demonstrates that our framework is both practical and efficient even under academic research constraints. We have included detailed discussions about costs in the revised manuscript.
>
>
>
> |    |   | Planner      |  |  | Writer  |  |   | Search API Call |     |
> | ---- | -- | --| -- | -- | --- | -- | --- | -- | --- |
> |      | Input token | Output token | Time  | Input token | Output token | Time  | Money of LLM API | Google Search   | ScraperAPI |
> | DeepResearch Bench   |  |   |   |   |  |  |  |   |  |
> | Gemini-2.5-pro-deepresearch    | -   | -  | -  | -   | -   | ~16 min  | -   | -  | -   |
> | WebWeaver (Claude-sonnet-4-20250514) | 196,808   | 10,298  | 13.4 min | 522,765     | 33,653  | 15.60 min | $2.82   | 20.24  | 112.25  |
> | WebWeaver (Qwen3-30b-a3b)   | 163,510     | 8,447        | 11.8 min | 564,676     | 36,619  | 6.0 min  | -                | 16.65           | 110.23     |
> | DeepConsult     |  |    |  |  |  | |  |  |  |
> | Gemini-2.5-pro-deepresearch      | -           | -            | -        | -  | -  | ~14 min  | -  | - | -  |
> | WebWeaver (Claude-sonnet-4-20250514) | 188,084     | 9,045  | 12.5 min | 471,929  | 29,525  | 14.8 min | $2.56 | 17.91 | 109.86  |
> | WebWeaver (Qwen3-30b-a3b)   | 151,381     | 7,933        | 10.1 min | 504,406     | 28,825 | 5.40 min | -  | 12.8  | 106.65     |
> | DeepResearch Gym   |  |      |  |  |  |  |  |  |  |
> | Gemini-2.5-pro-deepresearch  | -  | - | -   | -           | -            | ~11 min  | -   | -   | -          |
> | WebWeaver (Claude-sonnet-4-20250514) | 176,907     | 8,511        | 12.2 min | 504,341     | 33,698   | 14.4 min | $2.67  | 21.93           | 102.55     |
> | WebWeaver (Qwen3-30b-a3b)  | 117,091     | 6,767        | 9.30 min | 587,602     | 34,786   | 5.10 min | -  | 13.5  | 102 |
>
> Table R4. Cost analysis of different agent systems. The money costs ($) are computed based on OpenRouter pricing. The wall-clock time for `Gemini-2.5-pro-deepresearch` are manually measured from the Gemini web interface and represent the time for the final report generation. The token and money costs are unknown for `Gemini-2.5-pro-deepresearch`.

---

> ### Author Response · Authors · 2025-11-20
> **Response to Reviewer 4PPn (2/7)**
>
> > #### W2. Baseline comparability is unclear. WebWeaver is evaluated with multiple powerful base LLMs (Qwen-235B, Claude Sonnet 4), and even when the “agent model” is Claude, a separate model (GPT-oss-120B) is used for URL selection and evidence extraction. This multi-model, auxiliary-LLM design could materially boost performance yet is not mirrored for baselines. It is therefore difficult to isolate method gains from model capability/tooling differences.
>
>
>
> Thank you for this comment on ensuring fair comparability. To precisely isolate the gains attributable to our framework itself, separate from the underlying model's power, we conducted a new set of strictly controlled experiments. In these head-to-head comparisons, both WebWeaver and the widely-used ReAct paradigm baseline were equipped with the **identical backbone LLM (Qwen3-235b-a22b) and the same tools**. As shown in Table R5, WebWeaver significantly outperforms ReAct across all benchmarks (e.g., boosting the DeepResearch Bench score from 46.16 to 50.80). This provides direct evidence that the architectural design of WebWeaver—specifically its outline optimization and hierarchical structure—is the primary driver of its superior performance.
>
> Furthermore, to address your concern about the auxiliary LLM, we first clarify its role. For complex long-input, long-output tasks like open-ended deep research (OEDR), which require collecting information from numerous web pages, intermediate steps like URL selection and evidence extraction are not just optimizations but necessary engineering practices to manage the context length. To test whether our framework's success is critically dependent on the model used for this necessary step, we conducted an ablation study. We replaced the powerful GPT-oss-120b model with a smaller model (Qwen3-30b-a3b) for these tasks. The results in Table R6 show only a marginal performance drop (from 96.74 to 96.68 on DeepResearchGym), demonstrating that our framework is robust and not overly sensitive to the specific model used for the relatively low-complex evidence extraction tasks. This confirms that the overwhelming majority of the performance gain is attributable to our core architectural innovations.
>
>
>
> |   | DeepResearch Bench | DeepConsut | DeepResearchGym |
> | -- | --- | ---| --- |
> |     | Overall score  | Avg. score | Avg. score      |
> | ReAct (Qwen3-235b-a22b-instruct-2507)     | 46.16  | 5.68       | 86.72   |
> | WebWeaver (Qwen3-235b-a22b-instruct-2507) | **50.80** | **6.90**   | **95.57**  |
>
> Table R5. Controlled comparisons with the same LLM backbone and search tools.
>
>
>
> |    | DeepResearch Bench | DeepConsut | DeepResearchGym |
> | --- | ---- | --- | --- |
> | GPT-oss-120b  | 50.48  | 6.94 | 96.74  |
> | Qwen3-30b-a3b | 50.39 | 6.97  | 96.68  |
>
> Table R6. Ablation studies with varying summary models.
>
>
>
> > #### W3. Statistical analysis is thin.
>
>
>
> Thank you for this crucial feedback regarding the statistical analysis. We have conducted three full independent runs and have updated our results with means and standard deviation. The results, summarized in Table R7 below and detailed in our revised manuscript, demonstrate the high stability and reliability of our framework. For instance, the very low standard deviations for our top models (e.g., ±0.07 for Claude-sonnet-4 on DeepResearch Bench) confirm that even seemingly modest gains over strong baselines are consistent and statistically meaningful, not products of random chance.
>
> Furthermore, **please note that the evaluation scoring is based on a direct comparison against a reference answer written by a human expert, a score of 50 represents parity with this expert reference.** Our top model achieves 50.80, indicating that our reports are consistently judged as good as, or even superior to, the human standard.
>
>
>
> |  | | DeepResearch Bench |   | DeepConsut  | DeepResearchGym  |
> | --- | -- | --- | -- | -- | -- |
> |    | Overall score    | Eff. c.   | C. acc.  | Avg. score  | Avg. score       |
> | Gemini-2.5-pro-deepresearch  | 49.71  | 165.34  | 78.3  | 6.70  | 96.02  |
> | WebWeaver (Qwen3-30b-a3b-instruct-2507)   | 45.78 (0.68)     | 22.34 (2.58) | 25.29 (0.87)     | 3.99 (0.29)     | 77.61 (0.20)     |
> | WebWeaver (gpt-oss-120b)  | 47.71 (0.17)  | 66.83 (1.50) | 67.03 (0.87)     | 6.42 (0.08) | 92.28 (1.02) |
> | WebWeaver (Qwen3-235b-a22b-instruct-2507) | **50.80** (0.11) | 152.70 (6.34) | 75.72 (3.05) | 6.90 (0.14) | 95.57 (0.42) |
> | WebWeaver (Claude-sonnet-4-20250514) | 50.48 (0.07) | **216.99 (6.81)**  | **92.13 (0.70)** | **6.94 (0.02)** | **96.74 (0.10)** |
>
> Table R7. Benchmark performance of agent systems. Standard variance is present in bracket. Detailed comparison can be refered to Table 1 and 2 of the revised manuscript.

---

> ### Author Response · Authors · 2025-11-20
> **Response to Reviewer 4PPn (3/7)**
>
> > #### W4. Reliance on LLM-as-judge across all benchmarks creates evaluation circularity risks; although the paper follows “official” judges per benchmark, additional human evaluation on a representative subset would strengthen claims, especially for “insight,” “readability,” and “support.”
>
>
>
> Thank you for raising this comment. We have used the official evaluation metrics with the recommended judge LLMs of each benchmark in our manuscript. To complement the LLM-as-judge evaluations, we conduct a targeted human study for qualitative validation. We invite three PhD students with research experience for a blind, side-by-side comparison of our WebWeaver (Claude-sonnet-4) against the Gemini-2.5-pro-deepresearch baseline on a random subset of 15 samples from three used benchmarks. Evaluators were asked to determine a 'Win', 'Loss', or 'Tie' across six key dimensions: Instruction Following, Depth, Balance, Breadth, Support, and Readability.
>
> The results, summarized in Table R8, show a strong and consistent human preference for WebWeaver across all evaluated dimensions. Our system demonstrated particularly dominant performance in key areas, achieving a 76% win rate in Support (Failfulness) and over a 69% win rate in Depth (insightful). This study provides strong validation that aligns with our larger-scale LLM evaluations. The detailed criteria and human evaluation results have been added to our revised manuscript.
>
>
>
> |     |   | Instruction Following |  |  | Depth |  |   | Balance |   |   | Breadth |  |  | Support |   |    | Readability |   |
> | ---- | -- | -- | -- | -- | -- | -- | -- | --- | -- | --- | --- | -- | --- | -- | ---- | ---- | -- | --- |
> |   | Win  | Loss  | Tie  | Win  | Loss  | Tie  | Win  | Loss    | Tie  | Win  | Loss  | Tie  | Win  | Loss  | Tie  | Win  | Loss  | Tie  |
> | WebWeaver (Claude-sonnet-4) | 0.67 | 0.22  | 0.11 | 0.69 | 0.13  | 0.18 | 0.58 | 0.09    | 0.33 | 0.69 | 0.18 | 0.13 | 0.76 | 0.13    | 0.11 | 0.56 | 0.29 | 0.16 |
>
> Table R8. Human evaluation of WebWeaver (Claude-sonnet-4) and Gemini2.5-pro-deepresearch on random subsets (15 samples) of three benchmarks.
>
>
>
>
>
> > #### W5. Effective citations metric may be confounded by output length. WebWeaver outputs very long reports; if “Eff. c.” counts raw validated citations, longer outputs can inflate the score. There is no normalization for output length or per-claim basis reported. The very large Eff. c. numbers (200+) relative to baselines raise the possibility that the metric rewards verbosity rather than calibrated evidence use.
>
>
>
> Thank you for raising this comment about the relationship between output length and the officially provided Effective Citations (Eff. c.) metric. The high Eff. c. score is not a byproduct of verbosity, but a direct indicator of WebWeaver's superior ability to gather a comprehensive set of relevant documents and output a detailed, faithful report. Therefore, rather than an inflated score needing normalization, Eff. c. Can be viewed as the indirect metric for comprehensiveness and faithfulness.
>
> Furthermore, normalizing for output length by using a metric like "citation density" would be misleading and counterproductive. Such a metric would inadvertently penalize the very qualities of a high-quality report: in-depth analysis, nuanced explanation, and logical synthesis. A model optimized for density would be incentivized to produce short, disjointed claims merely to attach citations, while avoiding the essential "connective tissue" of reasoning that makes a report truly valuable.
>
> To properly evaluate calibrated use of evidence, we believe it is better measured by the citation accuracy. This ensures that each individual citation correctly and faithfully supports the specific claim it is cited. WebWeaver is designed to meet through its hierarchical retrieval and writing process. These two metrics demonstrate WebWeaver's ability to generate comprehensive and faithful reports.

---

> ### Author Response · Authors · 2025-11-20
> **Response to Reviewer 4PPn (4/7)**
>
> > #### W6. The ablation “hierarchical vs brute-force writing” lacks sufficient detail to be fully persuasive: context limits, chunking strategy for the brute-force baseline, and exact parity of planner outputs across conditions are not specified. Without these, it’s unclear if the baseline is disadvantaged by context overflow or suboptimal chunking rather than an inherent weakness of the paradigm.
>
>
>
> Thanks for your comment. The ablation “hierarchical vs brute-force writing” share the same conditions on the planning process but differ in the writing process. The main difference between hierarchical and brute-force writing are the context management. The brute-force writing strategy includes all the collected evidence from the planning process to generate the report. We do not chunk the collected evidence since they do not exceed 1M context length of Claude-sonnet-4. However, the extremely long input context still adversely affects the report quality and faithfulness. In contrast, our proposed hierarchical retrieval and writing strategy only provides essential and relavant evidence for each-step writing. The superior context management of WebWeaver results in the high-quality and faithful report generation. We have provided more details about the setting difference for this ablation study on the revised manuscript.
>
>
>
> > #### W7. The outline optimization ablation uses samples with three rounds of optimization and then evaluates earlier rounds; however, the prompt template prescribes “at least three” updates but the reported average outline optimization steps are ~2.2. This mismatch between prescribed behavior and observed behavior should be clarified.
>
>
>
> Thank you for pointing out this important detail. The instruction to perform "at least three" updates was designed as a strong heuristic to encourage the LLM to prioritize thorough refinement over premature completion. It sets a clear directive for the agent to iteratively improve its plan.
>
> However, the observed average of ~2.2 steps is not a failure to follow this directive, but rather a direct reflection of the agent's autonomous decision-making. For simpler queries where the initial outline and evidence were already deemed sufficient, the agent intelligently concluded that further refinement would yield diminishing returns. Consequently, it chose to select the "terminate" action early, which demonstrates the system's ability to dynamically balance a general instruction for thoroughness against the specific demands of the task.
>
>
>
> > #### W8. No code, no executable release, and no links to run-time infrastructure are provided. Prompts are included, but the system depends on multiple LLMs, tools, and scraping steps. Critical parameters (stop criteria, deduplication policies, cache behavior, search engine settings, rate limits, tie-breaking rules, etc.) are not fully specified. This reduces reproducibility.
>
>
>
> Thank you for the critical feedback on reproducibility. We are committed to full transparency and will open-source the entire WebWeaver framework upon acceptance of this paper. This includes the complete code for our planner, writer, tools, and memory system. We believe the code itself serves as the most definitive and executable specification, resolving any ambiguity about our implementation details.
>
> Regarding the WebWeaver-3k dataset, we have not yet decided when to release the static dataset. However, our open-source release will provide a more valuable capability: it will fully support the generation of complete trajectories and final reports from any customised queries. This empowers researchers to create their own data for tasks like model fine-tuning or distillation. To further aid this process, we have included the detailed methodology of our query collection pipeline in the revised manuscript for reference.

---

> ### Author Response · Authors · 2025-11-20
> **Response to Reviewer 4PPn (5/7)**
>
> > #### W9. The WebWeaver-3k dataset creation lacks overlap analysis against evaluation prompts and lacks licensing audits of scraped content. Without an overlap/contamination check, the risk of implicit test exposure cannot be discounted.
>
>
>
> Thank you for these critical questions regarding the integrity and creation of the WebWeaver-3k dataset. We address both data contamination and licensing below.
>
> **1. Overlap/Contamination Analysis**
>
> We crawled pages from public, high-quality sources like Reddit and then used a two-stage LLM-based process: 1) candidate extraction and 2) filtering and refinement based on strict criteria:
>
> - Open-endedness: Does the question require a detailed, synthesized report?
> - Clarity: Is the question well-posed and unambiguous?
> - Feasibility: Can evidence be found on the public web?
> - Refinement and Anonymization: If a question has potential, refine it to meet the above standards and remove any personally identifiable information.
>
> This yields queries that are more general compared to expert benchmarks that contains rich domain knowledge. To illustrate the fundamental stylistic difference:
>
> - An example from DeepResearch Bench: *"Research Topic: Crafting Techniques for Non-Alcoholic Cocktails. Objective: Investigate current non-alcoholic cocktails to discover innovative production methods and formulations."*
> - A representative example from WebWeaver-3k: *"What do you think about DeepSeek?"*
>
> Second, our dataset WebWeaver-3k is topically diverse and semantically different with benmarks. Its domain distribution is shown in Table R9, which covers ten domains. The Average maximum N-gram simiarlities between WebWeaver-3k and benchmarks are shown in Table R10. WebWeaver-3k achieves significantly low average maximum similarity (0.017 with DeepResearchGym when N=3), demonstrating that WebWeaver-3k does not induce any text exposure.
>
> **2. Licensing and Data Statement**
>
> We acknowledge the importance of ethical data handling. Our process included several safeguards:
>
> - Legal web access: we follow the website terms of service when crawling the pages.
> - Anonymization: As noted in our refinement criteria, all personally identifiable information (PII) was removed from the queries.
> - Only-Research Use: Our collection process was designed for non-commercial academic research and respected platform guidelines.
>
> We have improved the Ethic Statement of our revised manuscript for better data adherence of ICLR.
>
>
>
> |  | Technology & Science | Academics & Education | Entertainment | Economics & Business | Sports | Work & Life | Politics | Culture | Society | History |
> | --- | --- | --- | -- | --- | -- | -- | --- | -- | --- | -- |
> | Ratio | 23.9   | 16.6  | 16.4 | 15.4  | 7.8  | 5.9  | 4.7  | 3.4  | 3.2  | 2.3 |
>
> Table R9. Domain distribution of WeaWeaver-3k.
>
>
>
> |  | DeepResearch Bench | DeepConsult | DeepResearchGym |
> | -- | --- | -- | -- |
> | 2-Gram | 0.096  | 0.078  | 0.054 |
> | 3-Gram | 0.033  | 0.027 | 0.017 |
>
> Table R10. Average maximum N-gram similarity (N=2, 3) between WebWeaver-3k and benchmarks.
>
>
>
>
>
> > #### W10. The ethics section is a little generic at the moment and should also address website terms-of-service compliance, robots.txt adherence, or how PII in web pages is handled during evidence extraction and dataset curation. For an agent that crawls and stores web-scale evidence, these omissions are material.
>
>
>
> Thank you for raising these critical ethical considerations, which are paramount for any web-scale agent. We have expanded our ethics statement in the revised manuscript to explicitly address these points.
>
> This work adheres to the ICLR Code of Ethics. Our approach incorporates safeguards at multiple levels. First, our web crawler is designed to be a responsible web citizen: it respects `robots.txt` directives and uses standard, non-aggressive access patterns. Second, our data collection was conducted strictly for **non-commercial academic research**, aligning with the terms of service of the source platforms. Most importantly, we implemented a systematic safeguard for Personally Identifiable Information (PII). Our two-stage, LLM-based data curation process includes a mandatory "Refinement and Anonymization" step, which is specifically designed to remove any PII from the queries.

---

> ### Author Response · Authors · 2025-11-20
> **Response to Reviewer 4PPn (6/7)**
>
> > #### Q1. Code/data release: Will you release the full WebWeaver code (planner/writer, tools, memory) and the WebWeaver-3k dataset? If not, what components will be made public and when?
>
> ​	Please refer to the response to W8.
>
>
>
> > #### Q2. Fairness controls: Can you provide a controlled comparison where WebWeaver and a strong baseline agent both use the same backbone LLM and judge, to isolate framework gains? Any cross-judge robustness (e.g., alternate judges or human eval subsets)?
>
>
>
> Thank you for this suggestion. To precisely isolate the gains attributable to our framework, we have conducted a new set of controlled, end-to-end comparisons, with the results summarized in Table R11. In the first comparison, both WebWeaver and the ReAct baseline were equipped with the identical backbone LLM (Qwen3-235b), the same set of tools, and operated in the same environment. The only architectural difference was that WebWeaver employed its core **outline optimization** and **hierarchical retrieval and writing** modules, whereas ReAct followed a standard linear thought-action loop to collect information and output the final report. Similarly, we compared our framework against the Claude-research baseline, a proprietary powerful single-agent system, using the same Claude-sonnet-4 model to further validate our approach.
>
> The results clearly demonstrate the substantial benefits of the WebWeaver framework. As shown in Table R11, when using the exact same backbone LLM, WebWeaver consistently and significantly outperforms the baselines across all benchmarks. For instance, with Qwen3-235b, our framework boosts the DeepResearch Bench score from 46.16 to 50.80 and the DeepConsult score from 5.68 to 6.90. A similar significant uplift is observed that WebWeaver significantly outperforms Claude-research with the same Claude-sonnet-4. This provides strong and direct evidence that our performance improvements are not merely a function of the underlying LLM's power, but are directly attributable to the structural advantages conferred by our **outline optimization** and **hierarchical retrieval and writing** methods.
>
> For the human evaluation, please refer to the response to W4.
>
>
>
> |     | DeepResearch Bench | DeepConsut | DeepResearchGym |
> | -- | -- | -- | -- |
> |    | Overall score  | Avg. score | Avg. score      |
> | ReAct (Qwen3-235b-a22b-instruct-2507)  | 46.16  | 5.68 | 86.72           |
> | WebWeaver (Qwen3-235b-a22b-instruct-2507)  | **50.80** | **6.90**   | **95.57**  |
> | Claude-reserach (Claude-sonnet-4-20250514) | 45.0  | 4.60  | 80.25      |
> | WebWeaver (Claude-sonnet-4-20250514)  | **50.48**  | **6.94**   | **96.74** |
>
> Table R11. Controlled comparisons with varying LLM backbones.
>
>
>
>
>
> > #### Q3. Cost/latency: What are average wall-clock time, token counts, and dollar costs per task on each benchmark, for both planning and writing? How does it compare to baselines?
>
>
>
> Please refer to the response to W1.
>
>
>
>
>
> > #### Q4. Evidence extraction specifics: How are quote spans selected, how is evidence deduplicated/normalized, and what is the granularity for memory entries? How do you handle noisy, contradictory, or paywalled sources?
>
>
>
> Thank you for these detailed questions about our evidence extraction process. Our system directly uses a prompted LLM for both URL selection and evidence extraction. The LLM is instructed to select relevant URLs and extract evidence given a goal. The extracted evidence is then saved in the memory without any specific instruction on granularity. The exact prompts for these steps have been included in our revised manuscript.
>
> For handling problematic sources, our approach is straightforward. Paywalled or inaccessible sources are automatically identified  by the scraperAPI or may be timeout, so that we can skip them. Regarding noisy or contradictory evidence, we acknowledge this is a significant challenge. Our current system relies on the general synthesis capabilities of the writer LLM to handle minor inconsistencies. We have not implemented a specialized module for explicit contradiction detection, and we view this as a valuable direction for future research.

---

> ### Author Response · Authors · 2025-11-20
> **Response to Reviewer 4PPn (7/7)**
>
> > #### Q5. Failure and correction: How do the planner/writer detect and correct citation errors (e.g., broken links, drifted IDs, semantically mismatched evidence)? Any automatic validation?
>
>
>
> Thank you for this crucial question about citation integrity. Ensuring accurate and verifiable citations is a core design principle of WebWeaver. Our strategy is two-fold, focusing primarily on preventing citation errors through our system's architecture:
>
> **Consistent ID-Based Referencing:** To prevent errors like broken links or drifted IDs, we abstract away raw URLs from the LLM's direct handling. Upon retrieval, each unique source is assigned a simple, stable identifier (e.g.,  `<id_1>`). This ID is then used consistently across the planner, memory, and writer. This design choice is critical because LLMs are far more reliable at manipulating short, consistent IDs than at correctly reproducing long, complex URLs, thus mitigating the risk of hallucination or truncation errors during generation.
>
> **Hierarchical Retrieval and Writing:** This is our primary defense against **semantically mismatched evidence**. For each section of the final report, the writer is provided with only a small, curated subset of evidence and their corresponding IDs that are directly relevant to that specific sub-topic. By drastically reducing the "contextual noise" and the number of potential citations at each step, we minimize the risk of the model incorrectly associating a piece of evidence with the wrong source ID. It's a divide-and-conquer approach to citation.
>
>
>
> > #### Q6. WebWeaver-3k licensing and distribution: What is the licensing status of the dataset, given it is derived from web materials and proprietary teacher outputs? Can it be redistributed?
>
> Please refer to the response to W8.
>
>
>
> > #### Q7. Safety and misuse: What safeguards are in place to prevent the agent from citing low-quality or harmful sources? Do you filter by source credibility or domain allowlists/blacklists?
>
>
>
> Thank you for this question regarding safety and source quality. In our current framework, we have not implemented a specialized design to explicitly filter for low-quality or harmful sources, such as using domain allowlists, blacklists, or a source credibility scoring module. We agree that automatically determining the credibility, potential for harm, or bias of an arbitrary web source is a significant and open research challenge for the entire field. We consider this a highly valuable direction for future work and plan to dedicate effort to developing more robust safeguards within our framework.
>
>
>
> > #### Q8. Outline-to-retrieval mapping: How robust is the citation-ID mapping across multiple optimization rounds (IDs added/removed)? How do you maintain referential integrity?
>
> Thank you for this question. We ensure referential integrity through a simple yet robust mechanism: a centralized URL-to-ID mapping. Our system maintains a single dictionary for each query that maps every unique URL to a stable integer ID. When a URL is retrieved—at any step—the system checks this dictionary. If the URL is new, it is assigned a unique ID; if it has been seen before, its existing ID is reused. This ensures that a given URL has one and only one ID throughout the entire process. This ID is shared uniformly between the planner, the writer, and the memory, effectively preventing any citation errors or ID drift.

---

> > ### Comment · Reviewer_4PPn · 2025-11-20
> > **Thanks a lot for addressing my comments. I've increased my score to 8**
> >
> > Your responses mostly address my questions, and I don't have any major lingering concerns regarding the quality of this work. As such, I've increased my rating to 8. Thanks again for taking the time to address my questions in detail!

---

> > > ### Author Response · Authors · 2025-11-21
> > > **We are pleased to hear that our responses have addressed your questions.**
> > >
> > > Thank you for your follow-up and for the positive evaluation of our revised manuscript! We are delighted to hear that we have addressed your concerns.

---

### Official Review · Reviewer_21rJ · 2025-11-05

**Soundness:** 4
**Presentation:** 4
**Contribution:** 3
**Rating:** 8
**Confidence:** 4

**Summary:**

The paper introduces WebWeaver, a dual-agent framework for open-ended deep research (OEDR).
The system consists of a Planner that dynamically do evidence acquisition and outline optimization, and a Writer that performs memory-grounded, hierarchical synthesis by section. The human-centric design simulates real research workflows.
WebWeaver is claimed to achieve SOTA on DeepResearch Bench, DeepConsult, and DeepResearchGym against Gemini-2.5, OpenAI DeepResearch and Claude Research.

**Strengths:**

- Clear and well-motivated problem framing: tackling OEDR beyond static pipelines.
- Dual-agent design (Planner + Writer) inspired by real-wold human research process.
- Extensive experiments across 3 major benchmarks with detailed breakdowns with strong quantitative evidence.
- Thorough ablation and statistical analysis showing the strength of outline optimization and hierarchical writing.
- Clear figures, intuitive workflow, and solid writing throughout.

**Weaknesses:**

- Heavy rely on LLM-as-judge evaluations that lacks human verification to some extend
- I'm curious what's the computational cost and latency?
- Some redundancy between Sections 3.2 and 3.3, the method flow could be slightly more concise.
- How scalable is WebWeaver for real-world web-scale research (e.g., 1k+ page retrieval)?
- How sensitive is performance to the backbone model (Qwen vs. Claude vs. GPT)?
- Minor formatting issue: Fig. 1 move to page 2 looks better

**Questions:**

see weakness.

---

> ### Author Response · Authors · 2025-11-20
> **Response to Reviewer 21rJ (1/2)**
>
> > #### W1. Heavy rely on LLM-as-judge evaluations that lacks human verification to some extend
>
>
>
> Thank you for raising this comment. We have used the official evaluation metrics with the recommended judge LLMs of each benchmark in our manuscript. To complement the LLM-as-judge evaluations, we conduct a targeted human study for qualitative validation. We invite three PhD students with research experience for a blind, side-by-side comparison of our WebWeaver (Claude-sonnet-4) against the Gemini-2.5-pro-deepresearch baseline on a random subset of 15 samples from three used benchmarks. Evaluators were asked to determine a 'Win', 'Loss', or 'Tie' across six key dimensions: Instruction Following, Depth, Balance, Breadth, Support, and Readability.
>
> The results, summarized in Table R1, show a strong and consistent human preference for WebWeaver across all evaluated dimensions. Our system demonstrated particularly dominant performance in key areas, achieving a 76% win rate in Support (Failfulness) and over a 69% win rate in Depth. This study provides strong validation that aligns with our larger-scale LLM evaluations. The detailed criteria and human evaluation results have been added to our revised manuscript.
>
>
>
>
>
> | || Instruction Following | | | Depth | |  | Balance |  |  | Breadth |  |  | Support | || Readability | |
> | -- | - | - | -- | - | - | -- | -- | -- | -- | -- | -- | -- | -- | -- | -- | -- | -- | -- |
> |  | Win | Loss| Tie| Win | Loss | Tie | Win | Loss | Tie  | Win  | Loss | Tie | Win | Loss| Tie | Win  | Loss | Tie |
> | WebWeaver (Claude-sonnet-4) | 0.67 | 0.22 | 0.11 | 0.69 | 0.13  | 0.18 | 0.58 | 0.09 | 0.33 | 0.69 | 0.18 | 0.13 | 0.76 | 0.13 | 0.11 | 0.56 | 0.29 | 0.16 |
>
> Table R1. Human evaluation of WebWeaver (Claude-sonnet-4) and Gemini2.5-pro-deepresearch on random subsets (15 samples) of three benchmarks.
>
>
>
> > #### W2. I'm curious what's the computational cost and latency?
>
>
>
> We thank the reviewer for this important question regarding costs. We provide a detailed cost analysis in Table R2. We must first note that wall-clock times are highly sensitive to external factors like API latency, network conditions, and concurrent GPU capabilities (8 * H20 GPU). As our experiments were conducted under resource-constrained academic conditions, the reported times should be viewed as a practical reference rather than an absolute benchmark metric. With that context, WebWeaver with Claude-sonnet-4 spends more wall-clock time (e.g., ~29 min and $2.82 per task on DeepResearch Bench) compared with the strong baseline Gemini-2.5-pro-deepresearch. This higher cost is a direct consequence of its comprehensive evidence gathering, which enables the high-quality reports for writing.
>
> To provide a more efficient alternative, our fine-tuned model, Qwen3-30b-a3b, drastically reduces latency for planning and writing. Notably, the total end-to-end time ( \~18 min on DeepResearch Bench) is highly competitive with the Gemini baseline's estimated generation time (~16 min). This demonstrates that our framework is both practical and efficient even under academic research constraints. We have included detailed discussions about costs in the revised manuscript.
>
>
>
> |    |   | Planner |  |  | Writer|  | | Search API Call |  |
> | -- | -- | --| -- | -- | - | -- | -- | -- | --- |
> |  | Input token | Output token | Time | Input token | Output token | Time | Money of LLM API | Google Search | ScraperAPI |
> | DeepResearch Bench || | |  |  | | |  |  |
> | Gemini-2.5-pro-deepresearch | - | - | -  | - | - | ~16 min  | -   | -  | - |
> | WebWeaver (Claude-sonnet-4-20250514) | 196,808   | 10,298  | 13.4 min | 522,765 | 33,653  | 15.6 min | $2.82   | 20.24  | 112.25  |
> | WebWeaver (Qwen3-30b-a3b)   | 163,510 | 8,447  | 11.8 min | 564,676 | 36,619  | 6.00 min | -  | 16.65  | 110.23 |
> | DeepConsult  |  | |  |  |  | |  |  |  |
> | Gemini-2.5-pro-deepresearch | - | - | - | -  | -  | ~14 min  | -  | - | -  |
> | WebWeaver (Claude-sonnet-4-20250514) | 188,084| 9,045  | 12.5 min | 471,929  | 29,525  | 14.8 min | $2.56 | 17.91 | 109.86  |
> | WebWeaver (Qwen3-30b-a3b)   | 151,381 | 7,933  | 10.1 min | 504,406     | 28,825 | 5.40 min | -  | 12.8  | 106.65 |
> | DeepResearch Gym   |  | |  |  |  |  |  |  |  |
> | Gemini-2.5-pro-deepresearch  | -  | - | -   | -  | - | ~11 min  | -   | - | - |
> | WebWeaver (Claude-sonnet-4-20250514) | 176,907     | 8,511  | 12.2 min | 504,341 | 33,698   | 14.4 min | $2.67  | 21.93  | 102.55 |
> | WebWeaver (Qwen3-30b-a3b)  | 117,091  | 6,767 | 9.30 min | 587,602     | 34,786 | 5.10 min | -  | 13.5  | 102 |
>
> Table R2. Cost analysis of different agent systems. The money costs ($) are computed based on OpenRouter pricing. The wall-clock time for `Gemini-2.5-pro-deepresearch` are manually measured from the Gemini web interface and represent the time for the final report generation. The token and money costs are unknown for `Gemini-2.5-pro-deepresearch`.

---

> ### Author Response · Authors · 2025-11-20
> **Response to Reviewer 21rJ (2/2)**
>
> > #### W3. Some redundancy between Sections 3.2 and 3.3, the method flow could be slightly more concise.
>
> Thanks for your valuable suggestions. We have polished Section 3 for more concise in the revised manuscript.
>
>
>
>
>
> > #### W4. How scalable is WebWeaver for real-world web-scale research (e.g., 1k+ page retrieval)?
>
>
>
> Thank you for highlighting the importance of web-scale research. Our design is explicitly motivated by such complex problems.
>
> Current Scalability (>100 pages): We would like to highlight that WebWeaver is already operating at a high scale. In our experiments, the system retrieves over 100 pages on average, far exceeding the retrieval volume of standard baselines. This demonstrates its ability to filter and synthesize information from a massive pool of pages.
>
> Extending to 1k+ Pages: Regarding the specific 1,000+ page scenario, while current benchmarks do not yet demand this extreme scale, WebWeaver is architecturally ready. A naive, "brute-force" approach would fail catastrophically here. It would either suffer from massive context overflow or, even with a huge context window, its performance would degrade significantly due to the "lost in the middle" problem. In contrast, WebWeaver's hierarchical planning and writing is fundamentally built to handle this. We agree that pushing autonomous agents to robustly handle 1,000+ page research represents the next frontier, and we believe our hierarchical framework provides a strong and necessary foundation for this significant and exciting future research direction.
>
>
>
> > #### W5. How sensitive is performance to the backbone model (Qwen vs. Claude vs. GPT)?
>
>
>
> Thank you for this crucial feedback regarding the statistical analysis. We have conducted three independent runs and have updated our results with means and standard deviation. The results, summarized in Table R3 below and detailed in our revised manuscript, demonstrate the high stability and reliability of our framework. As model capability increases, performance variance systematically decreases and become more stable. For instance, our strongest model, Claude-sonnet-4, not only achieves top performance but also exhibits the tightest standard deviation (e.g., ±0.07 on DeepResearch Bench).
>
>
>
> |   |  | DeepResearch Bench |   | DeepConsut      | DeepResearchGym  |
> | ---- | - | -- | -- | -- | -- |
> |     | Overall score    | Eff. c.    | C. acc. | Avg. score      | Avg. score  |
> | Gemini-2.5-pro-deepresearch  | 49.71  | 165.34 | 78.3  | 6.7             | 96.02 |
> | WebWeaver (Qwen3-30b-a3b-instruct-2507)   | 45.78 (0.68)     | 22.34 (2.58)  | 25.29 (0.87)  | 3.99 (0.29)  | 77.61 (0.20)     |
> | WebWeaver (gpt-oss-120b) | 47.71 (0.17) | 66.83 (1.50)  | 67.03 (0.87)     | 6.42 (0.08)     | 92.28 (1.02) |
> | WebWeaver (Qwen3-235b-a22b-instruct-2507) | **50.80** (0.11) | 152.70 (6.34)      | 75.72 (3.05) | 6.90 (0.14)  | 95.57 (0.42)  |
> | WebWeaver (Claude-sonnet-4-20250514) | 50.48 (0.07) | **216.99 (6.81)**  | **92.13 (0.70)** | **6.94 (0.02)** | **96.74 (0.10)** |
>
> Table R3. Benchmark performance of agent systems. Standard deviation is present in bracket. Detailed comparison can be refered to Table 1 and 2 of the revised manuscript.
>
>
>
>
>
> > #### W6. Minor formatting issue: Fig. 1 move to page 2 looks better
>
> Thanks for your valuable suggestions.  We have moved Fig. 1 to page 2.

---

> > ### Comment · Reviewer_21rJ · 2025-11-22
> >
> > Thank you for your explaination!

---

### Author Response · Authors · 2025-11-20
**Response Summary**

Dear ACs and Reviewers,

We extend our deepest gratitude to the reviewers for their dedicated time and insightful feedback throughout the review process. We are particularly encouraged by their recognition of our work's **strong motivation** (@21rJ, @W3Xh), **novel human-inspired design** (@21rJ, @4PPn, @yb9V, @W3Xh), **well presentation** (@W3Xh), and **SOTA performance** (@21rJ, @4PPn, @W3Xh) validated through **extensive experiments** (@21rJ, @yb9V) on widely-used benchmarks.

The major clarifications are summarized as follows:

- **Human Evaluation:** We have conducted a new human evaluation, confirming the superior quality and coherence of WebWeaver's generated reports over baselines (@21rJ, @4PPn).
- **Cost Analysis:** We now provide a detailed cost analysis, demonstrating that WebWeaver's slightly increased computational cost is a deliberate trade-off for its significant gains in report quality and accuracy (@21rJ, @4PPn, @yb9V, @W3Xh).
- **Statistical Analysis:** To ensure the robustness of our results, we have performed statistical analysis across three independent runs, confirming the stability and significance of WebWeaver's performance gains (@21rJ, @4PPn).
- **Expanded Baselines:** We have expanded our experiments to include more strong baselines (including open-source systems and a strictly controlled ReAct setup), further solidifying WebWeaver's state-of-the-art performance (@4PPn, @yb9V, @W3Xh).
- **Controlled Ablation Studies**: New ablation studies have been added to isolate the architectural gains of our framework, proving that WebWeaver's core components—not just the underlying LLM—are key to its success (@4PPn, @yb9V).
- **Data & Ethics Statement:** We have significantly expanded our Ethics Statement to provide full transparency on our data processing, including details on PII anonymization, adherence to `robots.txt`, and terms of service, ensuring our work aligns with the highest ethical standards (@4PPn, @yb9V).

We provided point-by-point responses to each reviewer as below. We have also included all these modifications into our revised manuscript with blue text.

---

### Author Response · Authors · 2025-12-02
**Summary of Discussion Phase**

Dear ACs,

We extend our deepest gratitude to all the reviewers for their time and constructive guidance, which were instrumental in elevating the quality of this work. We are particularly encouraged by their initial recognition of our work's **strong motivation** (@21rJ, @W3Xh), **novel human-inspired design** (@21rJ, @4PPn, @yb9V, @W3Xh), **well presentation** (@W3Xh), and **SOTA performance** (@21rJ, @4PPn, @W3Xh) validated through **extensive experiments** (@21rJ, @yb9V) on widely-used benchmarks. We also sincerely thank the ACs for overseeing and facilitating this productive review process.

In response to the insightful feedback, we have made substantial revisions to strengthen the paper. We conducted a new human evaluation to confirm the superior quality of our generated reports and performed statistical analysis across multiple runs to validate the robustness of our results. To further solidify our claims, we expanded our experiments to include stronger baselines, added a detailed cost analysis, and conducted controlled ablation studies to isolate the architectural contributions of our framework. Finally, we significantly expanded our Data and Ethics Statement to ensure full transparency regarding data handling and PII anonymization.

We are pleased that our revisions and detailed responses successfully addressed the reviewers' initial concerns. **This positive outcome was further reflected by the evidence that two reviewers raised their scores from 6 to 8 (Reviewer 4PPn on Nov. 20; Reviewer W3Xh on Nov. 21) in the very early discussion phase prior to the data leak, improving their overall score configuration from [8, 6, 6, 6] to [8, 8, 8, 6]**. Crucially, all three engaged reviewers (21rJ, 4PPn, and W3Xh) have now explicitly confirmed that their concerns have been resolved. We believe this strong consensus demonstrates that the manuscript has been significantly strengthened through the review process.

In light of the substantial improvements and the positive consensus from the reviewers, we sincerely hope the ACs will give our paper the most favorable consideration.

---

### Meta-Review · Area_Chair_AJBE · 2026-01-07

**Summary:**

Reviewers were highly positive for this paper. Most reviewers belied that the problem (improving open-ended deep research) is clear and a solution will be valuable. The authors' proposed technique, based on a interleaved paired agents approach, is interesting and fairly natural. There are a solid amount of experiments with good and convincing results. Additionally, the reviewers liked the human inspiration for the deep research approach taken here.

Among weaknesses, there are the usual challenges that are faced with these types of systems (costs, which came up in each of the reviews, reliability of evaluation, particularly when using LLM-as-a-judge, and  comparisons against particular baselines.) The authors' responses were pretty convincing here---there is nothing out of the ordinary in terms of disadvantages.

**Reviewer Concerns:**

The rebuttals were quite comprehensive. I believe all of the results have been taken care of; in particular the concerns around cost and comparisons against baselines. I do not believe there is anything remaining here that is problematic to the paper.

**Reviewer Scores:**

I will describe each review in turn:
R 21rJ: this reviewer was highly positive on the paper, with only a few concerns. All of these are clearly addressed by the rebuttal, so that the reviewer would have likely maintained their (high) score.

R 4PPn: this reviewer was positive, but had a number of noted weaknesses and questions. Many of these were more exploratory questions rather than specific downsides (e.g., when will the code be released, how can we be sure to handle confounders like length when using judges). The authors have a long rebuttal that tackles each of the points. The major concerns are addressed effectively (costs and baselines, in particular). The reviewer (who already raised their score) would have maintained their updated score.

R yb9V: the reviewer was also positive, and had similar questions and concerns to the first two. Similarly the concerns were addressed. The reviewer would have likely maintained or increased their score.

R W3Xh: the reviewer was positive and had relatively few concerns. These are similar (cost questions and comparison design queries). Similarly all questions addressed; the reviewer would have likely maintained or improved their positive score.

---

### Decision · Program_Chairs · 2026-01-26

Accept (Poster)